# Mono- and biallelic variant effects on disease at biobank scale

H. O. Heyne[1,2,3,4,5✉], J. Karjalainen[1,5,6,7,8], K. J. Karczewski[1,5,6,7,8], S. M. Lemmelä[1,9], W. Zhou[1,5,6,7,8], FinnGen[*], A. S. Havulinna[1,9], M. Kurki[1,5,6,7,8], H. L. Rehm[5,7,8], A. Palotie[1,5,6,7,8,10] & M. J. Daly[1,5,6,7,8✉]

Identifying causal factors for Mendelian and common diseases is an ongoing challenge in medical genetics[1]. Population bottleneck events, such as those that occurred in the history of the Finnish population, enrich some homozygous variants to higher frequencies, which facilitates the identification of variants that cause diseases with recessive inheritance[2,3]. Here we examine the homozygous and heterozygous effects of 44,370 coding variants on 2,444 disease phenotypes using data from the nationwide electronic health records of 176,899 Finnish individuals. We find associations for homozygous genotypes across a broad spectrum of phenotypes, including known associations with retinal dystrophy and novel associations with adult-onset cataract and female infertility. Of the recessive disease associations that we identify, 13 out of 20 would have been missed by the additive model that is typically used in genome-wide association studies. We use these results to find many known Mendelian variants whose inheritance cannot be adequately described by a conventional definition of dominant or recessive. In particular, we find variants that are known to cause diseases with recessive inheritance with significant heterozygous phenotypic effects. Similarly, we find presumed benign variants with disease effects. Our results show how biobanks, particularly in founder populations, can broaden our understanding of complex dosage effects of Mendelian variants on disease.

Rare genetic variants with large effects on disease can have direct implications for the development of potential treatments[1]; however, studying their effects comprehensively requires large, broadly phenotyped samples[4]. Identifying variants that influence disease risk only in the homozygous state (recessive inheritance) is particularly challenging, as the square of variant frequencies means that the homozygous state is often exceedingly rare. By contrast, in populations that have encountered a recent reduction in population size, certain founder diseases with recessive inheritance are present at higher frequencies. The Finnish population has experienced such bottleneck events and has been, historically, relatively isolated from other European populations[5]. As a result, the Finnish population is characterized by higher rates of DNA stretches with a common origin[6,7] carrying particular sets of genetic variants. This leads to higher rates of homozygosity, and increases the chance occurrence of pathogenic variants in a homozygous state that lead to diseases with recessive inheritance. In consequence, there is an enrichment of 36 specific Mendelian genetic diseases such as congenital nephrotic syndrome, Finnish type (CNF)[8] in certain areas of Finland today that show mostly recessive inheritance. These 36 (at present) 'founder diseases' are referred to as the 'Finnish disease heritage'[2,9]. Analogous founder diseases in other populations include Tay-Sachs disease and Gaucher disease in Ashkenazi Jewish

individuals[10]; Hermansky–Pudlak syndrome in Puerto Rican individuals[11]; and autosomal recessive spastic ataxia of Charlevoix–Saguenay and Leigh syndrome, French Canadian type in French Canadian individuals[12,13]. Populations that have undergone recent bottlenecks are also characterized by an excess of mildly deleterious variants, which are derived from rare variants that stochastically increased in frequency after a bottleneck event[4]. This has been previously shown in Finland through an excess of potentially deleterious probable loss-of-function (pLoF) variants at lower to intermediate frequencies (around 0.5%–5%)[3,14]. The higher allele frequencies of deleterious founder variants increases the statistical power for detecting disease associations. Isolated populations have thus been successfully used to map disease genes for decades[5,15,16].

Genetic variants can have different effects on disease when in a monoallelic state (only one allele carries the variant; heterozygous) or a biallelic state (both alleles carry the variant; homozygous) (see Fig. 1). For common genetic variants, early genome-wide association studies (GWASs) found that additive models captured most genotype–phenotype associations, including those with non-additive (also called dominance) effects[17]. The vast majority of GWASs were therefore conducted with additive models only. By contrast, Mendelian disease variants are rarely described as additive (or equivalently, as

[1]Finnish Institute for Molecular Medicine (FIMM), University of Helsinki, Helsinki, Finland. [2]Digital Health Center, Hasso Plattner Institute for Digital Engineering, University of Potsdam, Potsdam, Germany. [3]Hasso Plattner Institute for Digital Health at Mount Sinai, Icahn School of Medicine at Mount Sinai, New York, NY, USA. [4]Department of Genetics and Genomic Sciences, Icahn School of Medicine at Mount Sinai, New York, NY, USA. [5]Program for Medical and Population Genetics, Broad Institute of MIT and Harvard, Cambridge, MA, USA. [6]Stanley Center for Psychiatric Research, Broad Institute of MIT and Harvard, Cambridge, MA, USA. [7]Analytic and Translational Genetics Unit, Massachusetts General Hospital, Boston, MA, USA. [8]Center for Genomic Medicine, Massachusetts General Hospital, Boston, MA, USA. [9]Finnish Institute for Health and Welfare, Helsinki, Finland. [10]Psychiatric and Neurodevelopmental Genetics Unit, Department of Psychiatry, Massachusetts General Hospital, Boston, MA, USA. *A list of authors and their affiliations appears at the end of the paper. ✉e-mail: henrike.heyne@hpi.de; mark.daly@helsinki.fi

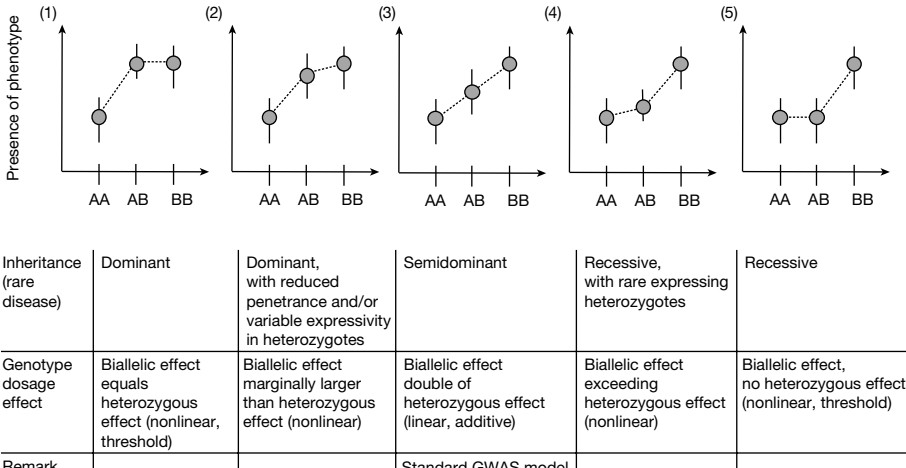

| Inheritance (rare disease) | Dominant | Dominant, with reduced penetrance and/or variable expressivity in heterozygotes | Semidominant | Recessive, with rare expressing heterozygotes | Recessive |
|---|---|---|---|---|---|
| Genotype dosage effect | Biallelic effect equals heterozygous effect (nonlinear, threshold) | Biallelic effect marginally larger than heterozygous effect (nonlinear) | Biallelic effect double of heterozygous effect (linear, additive) | Biallelic effect exceeding heterozygous effect (nonlinear) | Biallelic effect, no heterozygous effect (nonlinear, threshold) |
| Remark | | | Standard GWAS model | | |

**Fig. 1 | Schema of different effect sizes of monoallelic (heterozygous) versus biallelic variant states.** A is the wild-type and B is the mutant allele. We distinguish five main scenarios that are associated with different modes of inheritance used in rare-disease genetics (first row of the table at the bottom). In rare-disease genetics, the phenotypes associated with the mono- and the biallelic state in scenarios 2, 3 and 4 are usually viewed as distinct disease entities, with the monoallelic phenotype regarded as dominantly inherited and the biallelic phenotype, which is usually more severe, regarded as recessively inherited. In the schema, we focus on autosomal inheritance and do not show overdominant or underdominant inheritance (rare outside the HLA region). A perfectly linear additive genetic architecture (scenario 3) is also described, in which no dominance effect contributes to phenotypic variation.

semidominantly inherited[18,19]). Even for well-known examples of semi-dominant inheritance, such as the *LDLR* gene, which is associated with familial hypercholesteremia, diseases caused by monoallelic and biallelic variants are listed as separate recessive and dominant conditions in standard databases[20] (for example, ClinVar (https://www.ncbi.nlm.nih.gov/clinvar/) and the Online Mendelian Inheritance in Man (OMIM) database (https://www.omim.org/)). Although this nomenclature is limiting, for many Mendelian disease variants, monoallelic and biallelic phenotypes do have distinct features. For example, pathogenic variants in *ATM* cause cancer predisposition in a monoallelic and in a biallelic state, but ataxia telangiectasia only in a biallelic state. Genetic variants can also have distinct molecular effects that are differently inherited. In *SERPINA1*, for example, the same variant causes a dominantly inherited liver phenotype with a gain-of-function mechanism but a recessively inherited lung phenotype with a loss-of-function mechanism[21].

In this study, we analyse the effects of coding variants on 2,444 disease phenotypes using data from nationwide electronic health records of 176,899 Finnish individuals from the FinnGen project[22]. Participants are largely recruited through hospital biobanks and the data are thus enriched for individuals with diseases across the clinical spectrum. The phenotypes are derived from national healthcare registries collected over more than 50 years. In addition to the standard additive GWAS model[23], we systematically search for recessive associations. This enables us to examine two related questions of interest to both Mendelian and quantitative genetics communities. First, we investigate the potential benefit of searching for recessive associations in GWASs. Second, we consider the broad phenotypic consequences of Mendelian disease variants in the heterozygous as well as the homozygous state, and highlight how the current nomenclature could be improved to more precisely describe the complex inheritance of Mendelian variants.

## Recessive disease associations

In light of the global enrichment of deleterious variants in Finland[3,14] and the well-described Finnish disease heritage[2,9], we analysed data from the large population cohort gnomAD[24] and found that a set of variants that are known to cause disease with recessive inheritance are enriched in the Finnish population (Extended Data Fig. 1 and Supplementary Note 1). We thus saw the opportunity to identify novel recessive associations in our Finnish dataset. We performed a phenome-wide association study (pheWAS) to search for the effects of coding variants on 2,444 disease phenotypes in 176,899 Finnish individuals. When using the conventional GWAS model assuming additive genotype effects (see also scenario 3 in Fig. 1) in 82,647 variants, we found 1,788 significant associations ($P < 5 \times 10^{-8}$) for 445 coding variants in 305 genes[22]. For 44,370 coding variants with 5 or more homozygous individuals, we also investigated whether any of them had disease effects in the homozygous state (recessive GWAS model; see also scenario 5 in Fig. 1 and Supplementary Tables 1 and 2). We identified 124 associations—involving 39 unique variants—where the recessive model fitted substantially better than the additive model (recessive $P$ value two orders of magnitude smaller than the additive GWAS $P$ value; Fig. 2), corresponding to 31 unique loci (Fig. 2b, Supplementary Table 3 and Supplementary Fig. 1). Our simulations (Supplementary Note 2, Extended Data Fig. 2 and Supplementary Table 4) supported this as a suitable way to identify recessive associations. The simulations also showed how a recessive model can uncover associations that are missed by a conventional GWAS model—particularly for rare variants (minor allele frequency (MAF) < 0.05). This is consistent with more than half of our recessive associations not being genome-wide significant in the additive GWAS model (Table 1). Another 93 associations (of 73 unique variants) were identified in which the recessive $P$ value was smaller than the additive $P$ value (Supplementary Table 1). Those likely included non-additive effects as a simulated additive effect only showed a smaller recessive $P$ value than additive $P$ value in around 2 in 100,000 simulations.

Of the 31 unique loci, 13 were in known OMIM genes (Fig. 2), all of which were previously described with recessive disease inheritance. Of these, 9 of the 13 variants were known disease-causing variants[20] (6 pathogenic or likely to be pathogenic ('likely pathogenic'); 3 likely pathogenic by at least one submitter) and had large effects on disease, as expected. Among the remaining 18 variants in non-OMIM genes, we highlight *CASP7* (adult-onset cataract) and *EBAG9* (female infertility) (Fig. 3 and Supplementary Note 3). We investigated the effect of the *EBAG9* variant on fertility in 106,732 women born between 1925 and 1975 (thus with an approximately complete reproductive period) in our FinnGen data. We found that individuals homozygous for the *EBAG9* variant had fewer children ($1.7 \pm 1.3$ (mean ± s.d.)) than did wild-type individuals ($2.0 \pm 1.3$, Wilcoxon rank test $P = 0.0002$), and had their first child at a later age

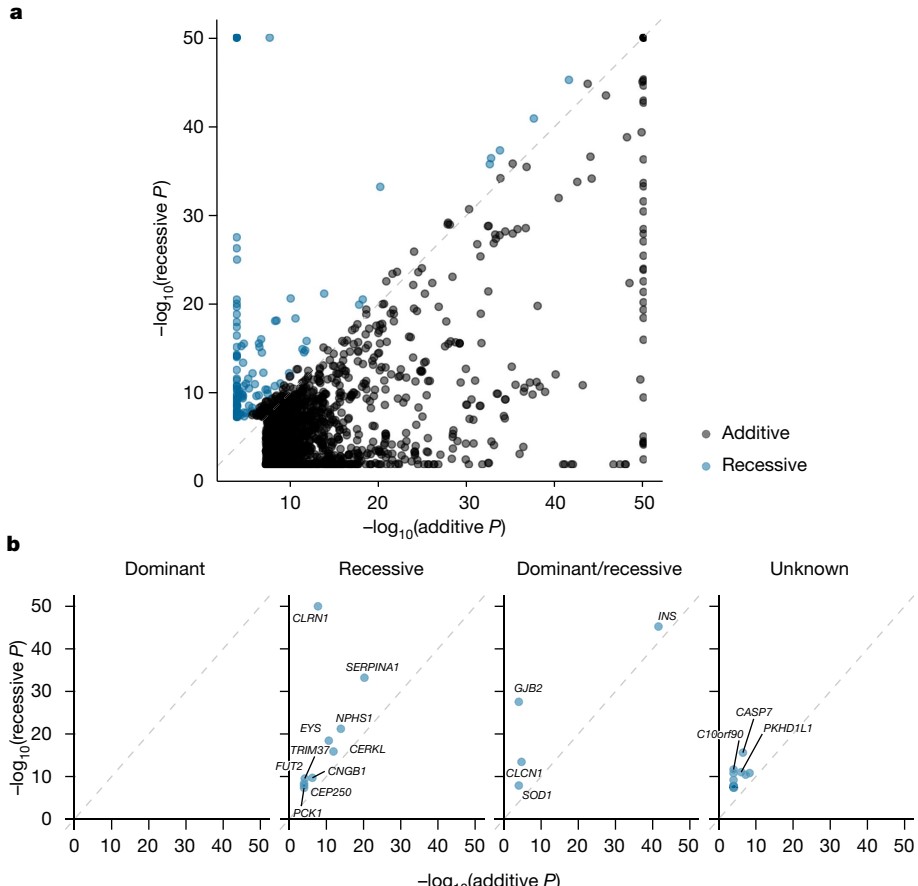

**Fig. 2 | *P* values of the additive versus the recessive GWAS model of all genome-wide-significant variant–disease associations.** Associations in which the *P* value of the recessive model is two orders of magnitude lower than that of the additive model are shown in blue (category: 'recessive'); all other associations are shown in black. **a**, All associations. **b**, Recessive associations, broken down by known inheritance modes of the respective disease gene (source: OMIM). Independent loci (considering adjacent variants with $r^2 > 0.25$ associated with the same (parent) trait as one locus). For clarity, $-\log_{10} P$ values are capped at 50.

(Extended Data Fig. 3), supporting the association of this variant with infertility. The six common (allele frequency greater than 10%) variants with recessive associations all overlapped known associations in the GWAS catalogue; however, for four of the six, a non-additive model of association had not been previously documented (Supplementary Table 5; variants in linkage disequilibrium with $r^2 > 0.1$; ref. [25]). Among the less-common variants, genes *GJB2* and *EYS* each contained two significantly associated likely pathogenic variants. For those and other genes, we thus identified compound heterozygous effects; that is, biallelic effects of two different pathogenic variants on separate alleles (see Extended Data Fig. 4 and Supplementary Note 4).

We next sought to validate any recessive associations in the UK Biobank (UKBB)[26] and in the FinnGen data freeze R6, an updated larger version of our original dataset. Owing to the isolation of the Finnish population[2], most of our variants with outsized homozygous effects were Finnish-enriched (allele frequency more than two times higher than that in Europeans who are not Finnish, Swedish or Estonian). Thus, in total, only 13 out of 31 variants were suitable for validation in the UKBB, 8 of which had nominally significant recessive associations to related phenotypes in the UKBB (Extended Data Fig. 5 and Supplementary Tables 6 and 7). We excluded seven variants with lower homozygote case counts in R6 compared to R4 as those may be due to technical differences between data freezes. Of the remaining 24 variants, 18 remained genome-wide significant in FinnGen R6. In total, we validated 20 recessive associations in FinnGen R6 and/or the UKBB (Table 1), including 4 novel associations. Recessive associations that could not be validated

are listed in Supplementary Tables 1 and 3. Most homozygous individuals also had a substantially earlier onset of disease than did wild-type individuals affected with the same diseases, providing additional evidence for the effect of these variants on disease (Extended Data Fig. 6).

We next investigated whether any of the variants with outsized homozygous effects also had subtle heterozygous effects on the same diseases, by excluding homozygous and compound heterozygous individuals from the analysis. We found variants with nominally significant heterozygous effects in *SERPINA1*, *NPHS1* and *CASP7* (see Table 1). In addition, we confirmed a previously hypothesized[27] heterozygous effect of a pLoF variant in *GJB2* on hearing loss in the larger R6 replication dataset (Fig. 4b; $P = 0.02$, $\beta = 0.11$). We thus classified the inheritance of these variants as recessive, with rare expressing heterozygotes (Fig. 1).

## Effects of known disease variants

Public databases of variants such as ClinVar[20], the largest, are central for routine clinical genetics but—as with many research community efforts—can include errors. We investigated whether pheWASs of 2,444 disease phenotypes could potentially enhance the interpretation of some ClinVar variants. Specifically, such efforts could support ClinVar submissions that are seen in individual patients with statistically robust associations. We first cross-validated our own results, and found the disease associations for the most frequent likely pathogenic variants across a wide range of phenotypes as expected (see Supplementary

## Table 1 | Recessive associations

| Gene | Lead variant (GRCh38) | OMiM | n hom. | Type | ClinVar | Phenotype | P value R4 | β R4 | P value R6 | β R6 | P value without hom. | β without hom. |
|---|---|---|---|---|---|---|---|---|---|---|---|---|
| *CLRN1* | 3:150928107:A:C | Reces. | 6 | pLoF | P/LP | Hereditary retinal dystrophy | $5.90\times10^{-16}$^ | 701.1 | $2.40\times10^{-18}$ | 648.3 | 0.57 | 0.48 |
| | | | | | | Sensorineural hearing loss | $2.50\times10^{-08}$ | 13 | $2.44\times10^{-09}$ | 15.2 | | |
| *CEP250* | 20:35504248:G:A | Reces. | 5 | Mis | | Varicose veins | $1.30\times10^{-08}$ | 4.3 | $2.23\times10^{-05}$ | 6.0 | 0.35 | −0.10 |
| *TRIM37* | 17:59079879:T:C | Reces. | 5 | pLoF | | Congenital malformation syndrome | $3.00\times10^{-10}$ | 37 | $1.80\times10^{-20}$ | 110.0 | 0.70 | −0.30 |
| *CERKL* | 2:181603943:G:C | NA | 10 | Mis | P/LP | Hereditary retinal dystrophy | $1.30\times10^{-16}$ | 210 | $1.32\times10^{-32}$ | 470.7 | 0.0035 | 2.70 |
| | | | | | | | | | | | (0.13)* | (0.9)* |
| *CNGB1* | 16:57901371:T:A | Reces. | 14 | Mis | Confl. | Hereditary retinal dystrophy | $2.00\times10^{-10}$ | 120 | $1.02\times10^{-22}$ | 359.7 | 0.11 | 1.19 |
| *EYS* | 6:63721375:TTCTGCATG:T | Reces. | 12 | pLoF | P/LP | Hereditary retinal dystrophy | $3.80\times10^{-19}$ | 220 | $1.07\times10^{-26}$ | 471.0 | 0.19 | 0.90 |
| *C10orf90* | 10:126459169:G:A | NA | 13 | Mis | B | Sensorineural hearing loss | $2.20\times10^{-12}$ | 6 | $8.83\times10^{-14}$ | 8.3 | 0.062 | 0.14 |
| *NPHS1* | 19:35851608:CAG:C | Reces. | 13 | pLoF | P | Nephrotic syndrome | $6.20\times10^{-22}$ | 34 | $9.85\times10^{-41}$ | 72.4 | 0.0046 | 1.2 |
| | | | | | | | | | | | (0.0004)* | (1.3)* |
| | | | | | | Glomerulonephritis | $1.30\times10^{-19}$ | 32 | | | | |
| *SOD1* | 21:31667290:A:C | Both | 9 | Mis | Confl. | Motor neuron disease | $1.30\times10^{-08}$ | 120 | $1.76\times10^{-19}$ | 214.0 | 0.17 | 0.82 |
| *PCK1* | 20:57563691:G:A | Reces. | 25 | Mis | Confl. | Glucose regulation and pancreatic secretion | $4.60\times10^{-08}$ | 9.8 | $1.80\times10^{-12}$ | 28.4 | 0.22 | −0.27 |
| *GJB2* | 13:20189546:AC:A | Both | 26 | pLoF | P | Sensorineural hearing loss | $2.90\times10^{-28}$ | 6.5 | $3.69\times10^{-42}$ | 14.7 | 0.28 | 0.07 |
| | | | | | | | | | | | (0.03)* | (0.12)* |
| *CASP7* | 10:113725526:T:C | NA | 45 | Mis | | Other cataract | $2.50\times10^{-16}$ | 4.8 | $3.95\times10^{-26}$ | 10.6 | 0.0035 | 0.25 |
| | | | | | | Senile cataract | $2.30\times10^{-12}$ | 2.1 | | | | |
| | | | | | | Medication related adverse effects (asthma or COPD) | $9.80\times10^{-11}$ | 1.5 | | | | |
| *CLCN1* | 7:143351678:C:T | Both | 53 | pLoF | Confl. | Diseases of myoneural junction and muscle | $3.70\times10^{-14}$ | 13 | $1.05\times10^{-15}$ | 20.0 | 0.20 | 0.21 |
| *SERPINA1* | 14:94378610:C:T | Reces. | 77 | | P | Emphysema | $2.20\times10^{-21}$ | 28 | $2.52\times10^{-31}$ | 51.8 | 0.037 | 0.5 |
| *EBAG9* | 8:109551075:C:G | NA | 339 | Intronic | | Female infertility | $1.60\times10^{-11}$ | 2 | $9.00\times10^{-16}$ | 1.98 | 0.08 | 0.09 |
| *TMEM214* | 2:27037601:G:A | NA | 14,561 | Mis | | Pain (limb, back, neck, head or abdomen) | $3.80\times10^{-11}$ | −0.061 | $6.05\times10^{-09}$ | −0.1 | 0.22 | 0 |
| *FUT2* | 19:48703417:G:A | Reces. | 2,5905 | pLoF | B | Intestinal infectious diseases | $4.00\times10^{-09}$ | −0.069 | $4.34\times10^{-15}$ | −0.2 | 0.06 | 0 |
| *IGHG3* | 14:105769806:G:A | NA | 29,078 | Mis | | Immunodeficiency with predominantly antibody defects | $4.40\times10^{-08}$ | 0.5 | $3.23\times10^{-05}$ | 0.6 | 0.45 | −0.1 |
| *UGT1A6* | 2:233693556:A:C | NA | 37,679 | Mis | B | Cholelithiasis | $1.50\times10^{-11}$ | 0.077 | $7.01\times10^{-23}$ | 0.2 | 0.31 | 0 |
| *INS* | 11:2159830:T:G | Both | 115,123 | Mis | B | Type 1 diabetes | $1.20\times10^{-41}$ | 0.24 | $9.95\times10^{-80}$ | 0.6 | 0.07 | 0.2 |

Reces., known recessive in OMIM; mis, missense; pLoF, probable loss-of-function; hom., homozygous individuals in FinnGen; P, pathogenic; LP, likely pathogenic; B, benign; Confl., conflicting evidence; COPD, chronic obstructive pulmonary disease; R4, FinnGen data release 4 (n =176,899); R6, FinnGen data release 6 (n=234,553). Only the most significant variant is shown per locus. Lead variants are given as rsids and GRCh38 coordinates (chromosome, position in bp, reference and alternate allele, separated by ':'). The P values and β values in brackets with an asterisk refer to the repetition of the heterozygous test in R6 data after exclusion of compound heterozygous individuals. The P value of the *CLRN1* variant's association with retinal dystrophy in R4 data (marked with ^) is calculated with Fisher's exact test as it did not converge in SAIGE with all homozygous individuals affected. *EBAG9* is the only noncoding variant presented here, as we estimate that it is a variant that is more likely to be causal than the originally identified pLoF variant in *PKHD1L1* (Supplementary Note 3). Variants in the bottom five rows (*TMEM214* and below) overlap with known GWAS loci that are listed in Supplementary Table 5.

Note 5, Supplementary Fig. 1 and Supplementary Table 8). In addition, we provide examples in which our data can help to verify or falsify previously described disease associations or identify novel associations (Fig. 4b). We then examined the collective association of several groups of ClinVar variants, as for many rare variants we were only powered to find their effects on disease with moderate significance (Methods

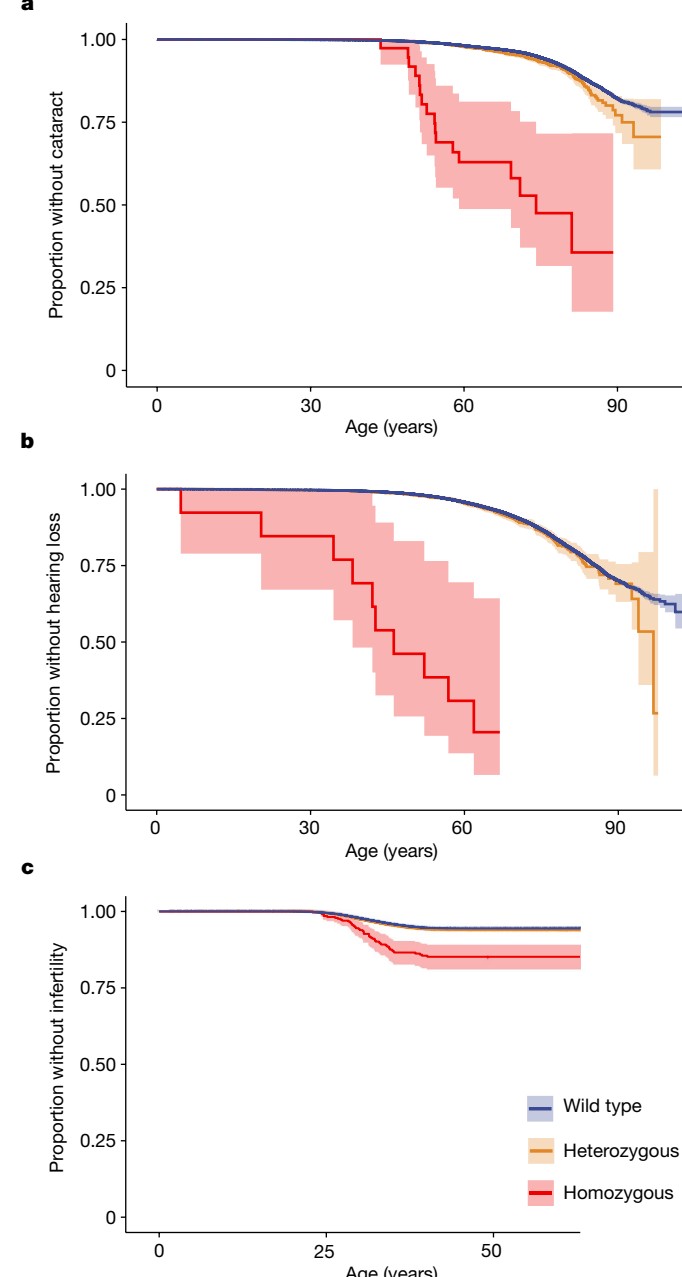

**Fig. 3 | Age at first diagnosis of variants with recessive disease associations.** Data are shown as survival plots. **a**, Missense variant in *CASP7* associated with cataract; not previously described ($P = 2.5 \times 10^{-16}$); $n = 176,899$. **b**, Missense variant in *C10orf90* associated with hearing loss ($P = 2.2 \times 10^{-12}$); only recently described[53]; $n = 176,899$. **c**, Intronic variant in *EBAG* associated with female infertility ($P = 1.6 \times 10^{-11}$); not previously described; $n = 110,361$ female individuals. Survival curves of wild-type individuals are coloured in blue, heterozygotes in yellow and homozygotes in red. The 95% confidence intervals of the point estimates are shaded in light blue, light yellow or light red.

and Extended Data Fig. 7). As anticipated, we found global disease associations for likely pathogenic variants in genes that are described to cause disease with dominant inheritance (classification: OMIM). However, we also found global disease associations for variants that are listed as benign or likely to be benign ('likely benign'). Likely benign variants are defined as 'not implicated in monogenic disease'[28], and are often considered to be neutral[29]. However, 16 likely benign variants were likely causally associated with disease in our data (using

statistical fine-mapping[30]; see Supplementary Table 9 and Supplementary Note 6). None of them had Mendelian effect sizes and would thus not cause monogenic disease. Rather, they moderately increased disease risk or protected from phenotypes that are mostly similar to the Mendelian phenotypes that are associated with the same gene. Of these benign variants, we highlight a variant in the gene *DBH*. *DBH* is a gene associated with dopamine beta-hydroxylase deficiency (inheritance: recessive), which is characterized by severe hypotension[31]. We observed a likely benign missense variant in *DBH* that conveyed protection from hypertension (see Fig. 4c), a plausible finding given the association of *DBH* with hypotension. The variant is also an example for a Finnish-enriched variant (allele frequency of 0.05, which is 22 times higher than that in Europeans who are not Finnish, Swedish or Estonian).

## Variants with monoallelic and biallelic effects

Because we observed unexpected disease effects from heterozygous variants in genes with reported recessive inheritance, we sought to further contrast the effects of variants with their previously described modes of inheritance. We found multiple coding variants with effects that did not match their annotated inheritance. These include known variants in *CHEK2*, *JAK2*, *TYR*, *OCA2* and *MC1R* (previously described as dominant inheritance) that have additive effects on cancer and cancer-related phenotypes (Supplementary Fig. 2). Similarly, we highlight one variant in *SCN5A* that was previously associated with severe cardiac-arrhythmia-like sick sinus syndrome[32] in a biallelic state, which we confirm in our data (Fisher's exact test, $P = 9 \times 10^{-4}$; odds ratio (OR) = 48 (95% confidence interval: 6–319)). In a heterozygous state, however, that same *SCN5A* variant protects from cardiac arrhythmia in FinnGen ($\beta = -0.48$, $P = 2 \times 10^{-8}$, posterior inclusion probability = 0.996 (ref. [30]), indicating probable causality), including atrial fibrillation ($\beta = -0.62$, $P = 7 \times 10^{-7}$). We could replicate this association in the UKBB[26] ($\beta = -0.39$, $P = 0.04$). In line with the subtle heterozygous effects of individual variants, we found global disease effects of 203 likely pathogenic variants in disease genes with recessive inheritance (see Extended Data Fig. 7 and Methods). Both simulations and reanalysis after excluding variants with homozygotes from the global analysis indicated that modest heterozygous effects are likely to contribute to this signal. In disease genes with known recessive and/or dominant inheritance, we found 79 additional coding variants (not likely pathogenic) with effects on disease that are likely to be additive (additive $P$ < recessive $P$) and causal[30]. Of these 79 variants, 11 had Mendelian effect sizes (OR > 3). In summary, we identify several variants that have disease effects in a heterozygous and a homozygous state in known Mendelian disease genes. The modes of inheritance of these variants cannot be described with the usual labels of recessive and dominant. Our data thus indicate the need for a nomenclature that integrates Mendelian and complex genetic effects on diseases. We outline a suggestion in Fig. 1.

## Discussion

A subset of disease-causing variants is enriched to unusually high frequencies in populations with a history of recent bottlenecks, such as the Finnish population. This applies particularly to homozygous variants that cause disease with recessive inheritance. The FinnGen cohort, given its size and its inclusion of broad medical phenotypic data over the lifespan of an individual, is well-powered for discovering novel alleles with large effects in homozygous individuals and for studying the disease architecture of Mendelian variants.

We found multiple variants in known Mendelian disease genes[20] with large effects in homozygous individuals that have weaker but significant effects in heterozygous individuals for the same or closely related diseases, highlighting that describing their inheritance with only dominant and recessive labels does not adequately describe disease biology. Terms beyond recessive and dominant are, however, rarely used in

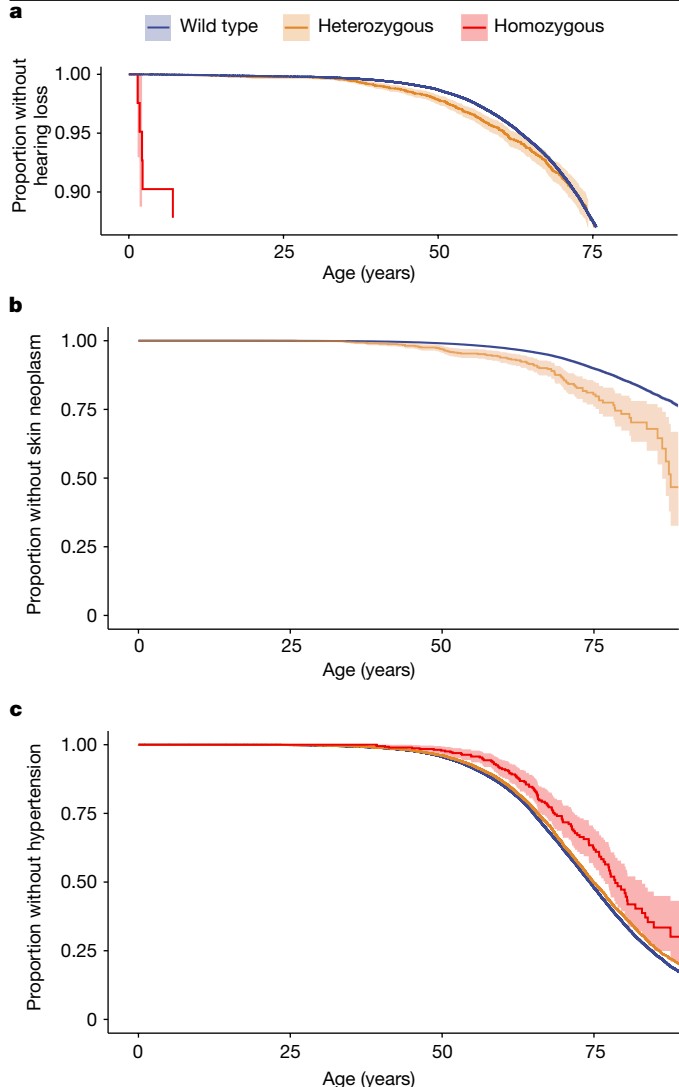

**Fig. 4 | Age at first diagnosis of known disease-associated variants.** Data are shown as survival plots. **a**, Known likely pathogenic variant (known recessive inheritance) in *GJB2* associated with hearing loss also in a heterozygous state (*P* = 0.02). The *y* axis is cut at 0.9 for clarity. **b**, Known likely pathogenic variant in *XPA* associated with skin cancer (*P* = 8 × 10⁻¹¹). In a homozygous state, this variant causes xeroderma pigmentosum with childhood-onset skin cancer[40]. **c**, Likely benign missense variant in *DBH* protects from hypertension (*P* = 5.2 × 10⁻¹³). (*DBH* is associated with the recessively inherited disease dopamine beta-hydroxylase deficiency, which is characterized by severe hypotension[31]). **a** and **b** show R4 data (*n* = 176,899); **c** shows R6 data (*n* = 234,553). Survival curves of wild-type individuals are coloured in blue, heterozygous individuals in yellow and homozygous individuals in red. The 95% confidence intervals of the point estimates are shaded in light blue, light yellow or light red.

clinical genetics. Although it has been estimated that semidominant inheritance is much more common than true dominant inheritance[33], phenotypes of homozygotes are frequently unknown because they are rare or too severe for an individual with the phenotype to survive to birth. The term semidominant inheritance is more frequently used in animal and plant genetics[34], in which mono- and biallelic effects can be more easily quantified and systematically studied. Aggregating the evidence for pathogenicity across biallelic and monoallelic observations, rather than viewing these as separate disease entities, could benefit the interpretation of clinical variants, as well as providing a

more accurate description of biology. Furthermore, we found that several variants that were previously described as likely to be benign are associated with disease. This is a noteworthy reminder that such likely benign variants are generally defined as not causing monogenic disease and should not necessarily be regarded as neutral[29].

Adding to an increased complexity of Mendelian inheritance, we detected modest heterozygous effects on disease in variants that are known to cause disease only in a homozygous state (recessive inheritance). This is in line with previous studies that found subtle heterozygous fitness effects of variants that are described as causing disease with recessive inheritance in mice[35], *Drosophila*[36] and humans[37]. We found heterozygous effects of such variants in the genes *NPHS1* (previously debated[38]), *SERPINA1* (known[21]) and *GJB2* (previously hypothesized[27]). As expected for recessive inheritance, the variants had an order-of-magnitude-larger effect size and, often, an earlier onset of disease in a homozygous than in a heterozygous state, thereby exceeding a linear additive model. We also found a heterozygous pLoF variant in *XPA* that increases susceptibility to adult-onset skin cancer. The effect was far larger than is usually found by GWASs and could thus provide valuable information on personal risk. Although a nominally significant association (*P* = 0.01) of a heterozygous pLoF variant in *XPA* with skin cancer was found in a previous study[39], we provide here the first—to our knowledge—definitive evidence. Homozygous pLoF variants in the gene *XPA* are known to cause a related phenotype, xeroderma pigmentosum, which is characterized by extreme vulnerability to UV radiation and childhood-onset skin cancer[40]. Long before these large-scale data became available, small heterozygous effects were found in variants that cause Mendelian disease with recessive inheritance[41], in some cases conferring an advantage against certain infectious diseases[42–44]. Similarly, we find that one variant in *SCN5A*, which was previously associated with severe cardiac arrhythmia such as sick sinus syndrome[32] in a biallelic state, protected from mild cardiac arrhythmia diseases in a heterozygous state in our data. Previous experimental data found a mild loss-of-function effect of this variant[32,45]. This is in line with a potential slowing of electrical conduction in the heart in a few individuals who are heterozygous for the *SCN5A* variant[32], which could thus provide a protective mechanism against cardiac arrhythmia.

Of course, it is possible that this and other heterozygous effects we observe come from low-frequency variants in a compound heterozygous state that were not captured by our genotyping array. However, a different age of disease onset in heterozygous than in homozygous individuals suggests that that scenario is unlikely to explain most of the observed heterozygous effect. In addition, with population-specific exome sequencing and imputation we could account for the presence of additional pathogenic variants at frequencies greater than 0.2%. Additional limitations are the lack of more in-depth phenotypes, including symptoms that are not captured by International Classification of Diseases (ICD) codes, or serological or diagnostic tests missing subtle physiological differences between heterozygous and wild-type individuals. Furthermore, the Finnish population bottleneck leads to a lower number of rare variants, which limits new discoveries. We cannot thus investigate relatively common European Mendelian disease variants; for example, variants in *CFTR*, which cause cystic fibrosis[46].

We systematically investigated recessive associations of coding variants genome-wide with pheWAS at biobank scale. We could validate 20 loci (4 of them novel) that had large biallelic effects without, or with only nominally significant heterozygous effects. Novel associations included complex non-syndromic diseases with few or no previously described large-effect variants, such as adult-onset cataract (new disease gene: *CASP7*) and female infertility (new disease gene: *EBAG9*). Intuitively, these novel findings appeared in phenotypes (deafness, cataracts and infertility) for which Mendelian subtypes would not obviously be clinically distinguished from other common presentations, although—as in previous examples—an earlier age of onset in the cases of common

diseases of ageing is seen. Biallelic associations of rare coding variants have been found in other population biobanks[47–51], but have not (to our knowledge) been investigated in a broad phenotype context in most studies, which lack the scale, the advantage of isolation and/or the high rates of homozygosity by descent. We suggest that searching for homozygous effects is most meaningful for coding and structural variants as opposed to regulatory variants with only weak effects. This is consistent with the observations that dominance effects are very modest in GWASs of common variants in complex human traits[52], but that recessive inheritance in Mendelian disease is widespread.

In summary, our biobank-scale additive and recessive pheWAS of coding variants shows the benefit of including recessive scans in GWASs. We find known and novel biallelic associations across a broad spectrum of phenotypes such as retinal dystrophy, adult-onset cataract and female infertility that are missed by the standard additive GWAS model. As a related point, we find an underappreciated complexity of inheritance patterns of multiple Mendelian variants. Our study could thus provide a starting point for reconciling the variant-effect nomenclature of the conventionally separate but more-and-more overlapping fields of Mendelian and complex genetics.

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

**FinnGen**

**H. O. Heyne[1,2,3,4,5], J. Karjalainen[1,5,6,7,8], S. M. Lemmelä[1,9], W. Zhou[1,5,6,7,8], A. S. Havulinna[1,9], M. Kurki[1,5,6,7,8], A. Palotie[1,5,6,7,8,10] & M. J. Daly[1,5,6,7,8]**

## Methods

### Ethics and data access approvals

The Coordinating Ethics Committee of the Hospital District of Helsinki and Uusimaa (HUS) approved the FinnGen study protocol HUS/990/2017. The FinnGen project is approved by the Finnish Institute for Health and Welfare (THL) (approval number THL/2031/6.02.00/2017; amendments THL/1101/5.05.00/2017, THL/341/6.02.00/2018, THL/2222/6.02.00/2018 and THL/283/6.02.00/2019), the Digital and Population Data Services Agency (VRK43431/2017-3 and VRK/6909/2018-3), the Social Insurance Institution (KELA) (KELA 58/522/2017, KELA 131/522/2018 and KELA 70/522/2019) and Statistics Finland TK-53-1041-17. The Biobank access decisions for FinnGen samples and data used in FinnGen data freeze 4 include: THL Biobank BB2017_55, BB2017_111, BB2018_19, BB_2018_34, BB_2018_67, BB2018_71 and BB2019_7, Finnish Red Cross Blood Service Biobank 7.12.2017, Helsinki Biobank HUS/359/2017, Auria Biobank AB17-5154, Biobank Borealis of Northern Finland_2017_1013, Biobank of Eastern Finland 1186/2018, Finnish Clinical Biobank Tampere MH0004, Central Finland Biobank 1–2017 and Terveystalo Biobank STB 2018001.

Patients and control individuals in FinnGen provided informed consent for biobank research, according to the Finnish Biobank Act. Alternatively, separate research cohorts, collected before the Finnish Biobank Act came into effect (in September 2013) and before the start of FinnGen (August 2017) were collected on the basis of study-specific consents and later transferred to the Finnish biobanks after approval by Fimea (Finnish Medicines Agency), the National Supervisory Authority for Welfare and Health. Recruitment protocols followed the biobank protocols approved by Fimea. The Coordinating Ethics Committee of the Hospital District of Helsinki and Uusimaa (HUS) statement number for the FinnGen study is HUS/990/2017. UK biobank data were accessed under protocol 31063.

### Funding and partners

We acknowledge the participants and investigators of the FinnGen study. The FinnGen project is funded by two grants from Business Finland (HUS 4685/31/2016 and UH 4386/31/2016) and the following industry partners: AbbVie, AstraZeneca UK, Biogen MA, Bristol Myers Squibb (and Celgene Corporation & Celgene International II Sàrl), Genentech, Merck Sharp & Dohme, Pfizer, GlaxoSmithKline Intellectual Property Development, Sanofi US Services, Maze Therapeutics, Janssen Biotech, Novartis and Boehringer Ingelheim International. The following biobanks are acknowledged for delivering biobank samples to FinnGen: Arctic Biobank (https://www.oulu.fi/medicine/node/207208), Auria Biobank (https://www.auria.fi/biopankki) THL Biobank (https://thl.fi/en/web/thl-biobank), Helsinki Biobank (https://www.helsinginbiopankki.fi), Biobank Borealis of Northern Finland (https://www.ppshp.fi/Tutkimus-ja-opetus/Biopankki/Pages/Biobank-Borealis-briefly-in-English.aspx), Finnish Clinical Biobank Tampere (https://www.tays.fi/en-US/Research_and_development/Finnish_Clinical_Biobank_Tampere), Biobank of Eastern Finland (https://ita-suomenbiopankki.fi/en/), Central Finland Biobank (https://www.sairaalanova.fi/en-US), Finnish Red Cross Blood Service Biobank (https://www.veripalvelu.fi/verenluovutus/biopankkitoiminta), Terveystalo Biobank (https://www.terveystalo.com/fi/yhtio/biopankki) and the Finnish Hematology Registry and Clinical Biobank (https://www.fhrb.fi). All Finnish Biobanks are members of the BBMRI infrastructure (https://www.bbmri-eric.eu/national-nodes/finland/). Finnish Biobank Cooperative (FINBB; https://finbb.fi) is the coordinator of BBMRI-ERIC operations in Finland.

### Coding variants in FinnGen, release 4

Greater haplotype sharing in the Finnish population facilitates the imputation of lower-frequency variants in array data with a population-specific reference panel down to frequencies below 0.0005 (ref. [54]). Genotypes were thus generated with arrays thereby enabling the large scale of the FinnGen research project. In summary, we investigated 82,647 coding variants (2,634 pLoF, 76,884 missense and 3,129 others) in the FinnGen project, release 4, 8/2019 in 176,899 Finnish individuals. We excluded the HLA region (chr. 6, 25 Mb–35 Mb). In 110,361 individuals, sex was imputed as female. We filtered to variants with INFO > 0.8 at a median allele frequency of 0.004 (minimum allele frequency $3 \times 10^{-5}$). Around half of the samples came from existing legacy collections, and the other half was from participants who were newly recruited to the FinnGen project. Samples were genotyped on custom microarrays and rare variants were imputed using a population-specific reference panel[54]. To calculate variant enrichment in Finnish individuals after a bottleneck event, we used as a general European reference point exomes from European samples in gnomAD 2.1.1, excluding those from Finland, Sweden and Estonia. Owing to large-scale migrations from Finland to Sweden in the 20th century, a substantial fraction of the genetic ancestry in Sweden is of recent Finnish origin, and the linguistically (and geographically) close population of Estonia is likely to share elements of the same ancestral founder effect. See ref. [22] or https://finngen.gitbook.io/documentation/ for a detailed description of data production and analysis.

### GWAS searching for additive and recessive associations

We performed a GWAS on 2,444 disease end-points, investigating the effects of 82,647 coding variants with an additive and recessive model using the method SAIGE[23]. Covariates in FinnGen were age, sex, genotyping batch and the first 10 principal components of genotypes. To identify heterozygous effects, we performed a GWAS with an additive model after excluding homozygous and compound heterozygous individuals, where possible (see Supplementary Note 4). In the recessive GWAS model we analysed the effects of homozygous alleles on disease phenotypes in comparison to wild-type and heterozygous alleles. The wild type is defined as the reference allele in FinnGen. The docker container finngen/saige:0.39.1.fg with all necessary software used to run SAIGE in additive or recessive mode can be found at the docker container library hub.docker.com. We replicated our genome-wide significant recessive associations in 234,553 individuals of FinnGen data freeze R6 and in 420,531 individuals with European ancestry in the UKBB using the same recessive model in SAIGE. GWAS covariates in UKBB were age, sex, age × sex, age$^2$, age$^2$ × sex and the first 10 principal components of genotypes. The genotype and phenotype files along with ancestry definitions, phenotype definitions and SAIGE null models were taken from the PAN UKBB project and are further described at https://pan.ukbb.broadinstitute.org. For additional information on and results from FinnGen data freeze R6 please search the indicated websites in the 'Data availability' section.

### Annotating variant effects from ClinVar

We annotated variants from release 25 March 2020 of ClinVar[20]. For any variant included in the main tables, we rechecked current classifications in ClinVar and OMIM on 2 November 2021. We grouped variants into categories according to their 'ClinVar_ReviewStatus'. Our main categories were likely pathogenic (likely to be pathogenic or pathogenic; 311 variants), conflicting evidence (at least one submitter labelled a variant as likely pathogenic but at least one other submitter labelled it different from likely pathogenic; 298 variants) and likely benign (likely to be benign or benign; 10,948 variants). The ClinVar annotation labelling variants as likely benign was above average quality (1.6; 7 of the 16 likely benign top causal variants had a one-star and 9 of 16 a two-star review status in ClinVar) compared to an average 1.2 stars for all likely benign variants in ClinVar (range: zero to three stars). Other categories into which we grouped variants that we are not explicitly discussing in the manuscript were 'association' (26 variants), 'drug response' (59 variants), 'not provided' (141 variants), 'protective' (14 variants), 'risk

factor' (74 variants) and 'VUS' (variant of unknown significance; 3,269 variants); see Supplementary Table 8.

## Annotating inheritance mode from OMIM
We downloaded the OMIM catalogue of human genetic diseases (https://www.omim.org/) version 06/2019. From OMIM, we annotated genes implicated to cause disease with a recessive or dominant inheritance mode.

## Global phenotype associations of ClinVar variant categories
We compared global disease phenotype associations of different ClinVar variant categories (likely benign, likely pathogenic or conflicting variants in genes with dominant or recessive inheritance) with phenotype associations of random intergenic variants. For a given variant category, we counted how many variants had at least one significant GWAS hit (2,444 phenotypes) below a given $P$ value threshold. We then compared those to the number of top GWAS loci below the $P$ value threshold of 1,000 random samples of intergenic variants. We calculated with empirical $P$ values if any ClinVar variant categories had significantly more disease associations than random intergenic variants below respective $P$ value thresholds. Allele frequency influences the power with which significant associations are identified. Therefore, we adjusted for allele frequency by sampling intergenic variants in 15 equal-sized bins that corresponded to the allele frequency of the variants under investigation. To account for linkage disequilibrium, we sampled intergenic variants from the same 3-Mb windows as variants in the respective gene set.

## Age at first diagnosis
We compared age at first diagnosis of homozygous or heterozygous individuals compared to wild-type individuals, respectively, using Wilcoxon rank tests. For a few compound heterozygous variants (as indicated in the paper), we also performed survival analyses using age at first disease diagnosis as outcome using a Cox proportional hazard model with the same covariates that were also used in the GWAS (sex, age, genotyping batch and first 10 principal components).

## Reporting summary
Further information on research design is available in the Nature Portfolio Reporting Summary linked to this article.

## Data availability
All summary statistics described in this manuscript can be found in the Supplementary Information. All information for downloading the summary statistics of additive GWASs of FinnGen release 4 can be found at https://finngen.gitbook.io/documentation/v/r4/data-download. Information for downloading the FinnGen release 6 summary statistics can be found at https://finngen.gitbook.io/documentation/v/r6/data-download. You can learn more about accessing other FinnGen data here: https://www.finngen.fi/en/access_results. A full list of FinnGen endpoints for release 4 and release 6 is available at https://www.finngen.fi/en/researchers/clinical-endpoints. A table of coding variants' summary statistics (any additive association with $P$ value $< 1 \times 10^{-4}$) has been included as a Supplementary Table 10. Individual-level genotypes and register data from FinnGen participants can be accessed by approved researchers through the Fingenious portal (https://site.fingenious.fi/en/) hosted by the Finnish Biobank Cooperative FinBB (https://finbb.fi/en/). Data release to FinBB is timed to the biannual public release of FinnGen summary results, which occurs 12 months after members of the FinnGen consortium can start working with the data.

## Code availability
Please see https://finngen.gitbook.io/documentation/ for a detailed description of data production and analysis, including the code that was used to run analyses, and https://github.com/FINNGEN/ for further code repositories that were used to run analyses in FinnGen. R code to reproduce figures is available upon request.

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

**Acknowledgements** We thank all FinnGen participants for their contributions to the research. Patients and control individuals in FinnGen provided informed consent for biobank research, on the basis of the Finnish Biobank Act. Research cohorts collected before the Finnish Biobank Act were collected on the basis of study-specific consents and later transferred to the Finnish biobanks after approval by Fimea, the National Supervisory Authority for Welfare and Health. The FinnGen project, data release 4, was funded by Business Finland and 13 industry partners. Further information on funding and approvals can be found in 'Funding and partners' in the Methods and the full list of members of the FinnGen consortium is provided in the Supplementary Information. We thank L. Biesecker, A. L. George, J. Kosmicki and J. Krause for discussions and comments.

**Author contributions** H.O.H. wrote the manuscript and generated the figures. H.O.H. and J.K. performed analyses. K.J.K., S.M.L., A.S.H., W.Z., M.K. and H.L.R. contributed data and research tools. A.P. and M.J.D. supervised the study. H.O.H. and M.J.D. conceived the study. All authors listed under FinnGen contributed to the generation of the primary data of the FinnGen data release 4. All authors reviewed the manuscript.

**Funding** Open access funding provided by University of Helsinki including Helsinki University Central Hospital.

**Competing interests** A.P. is a member of the Pfizer Genetics Scientific Advisory Panel. M.J.D. is a founder of Maze Therapeutics. The remaining authors declare no competing interests.

**Additional information**
**Correspondence and requests for materials** should be addressed to H. O. Heyne or M. J. Daly.

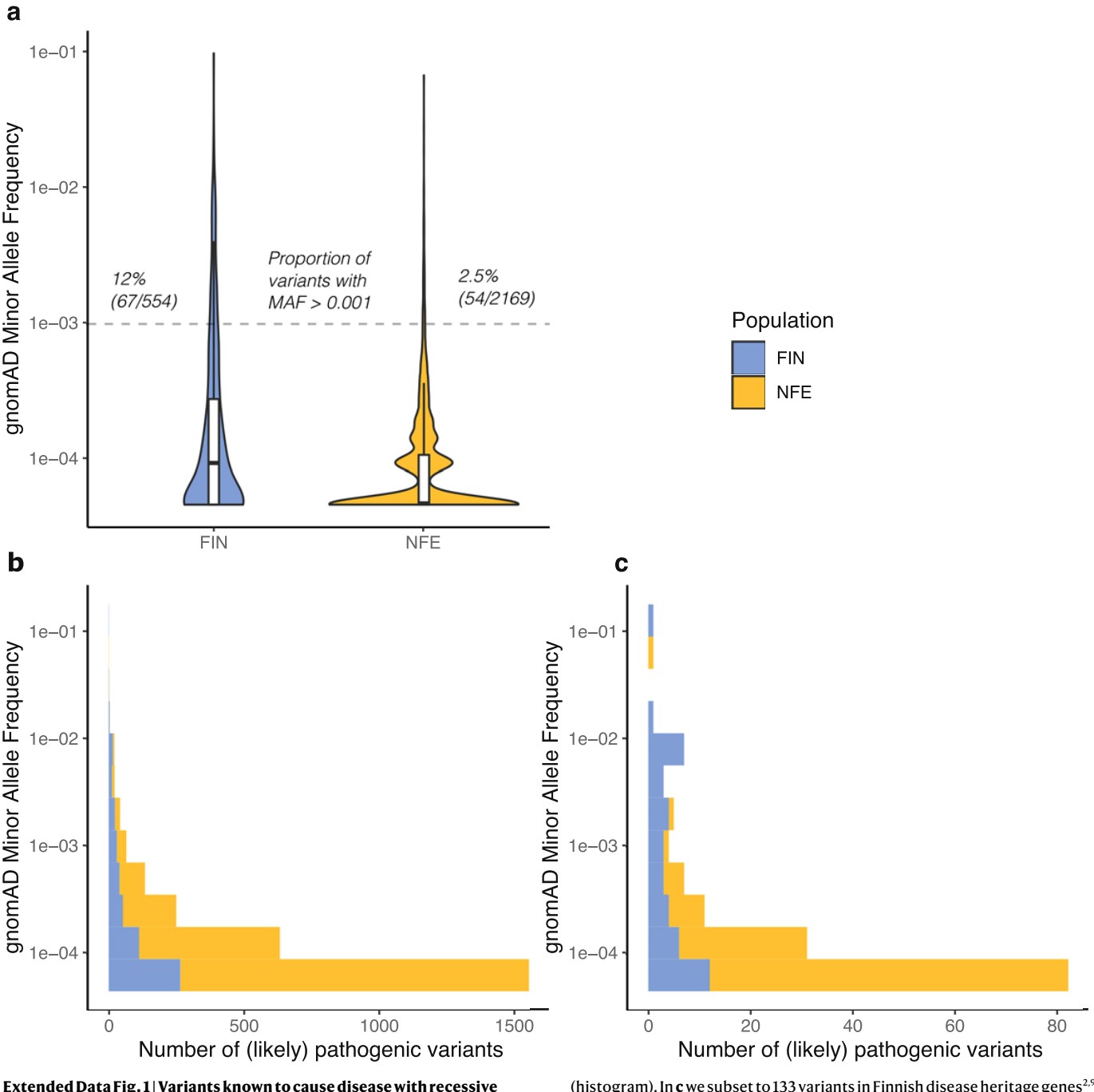

**Extended Data Fig. 1 | Variants known to cause disease with recessive inheritance are found at a higher MAF in Finnish Europeans than in other Europeans.** We show MAF of 2,419 unique likely pathogenic variants (source: ClinVar[20]) in 10,824 individuals from FIN and 10,824 from NFE populations (source: gnomAD[24]). **a**, All 2,419 variants (violin plot). **b**, All 2,419 variants (histogram). In **c** we subset to 133 variants in Finnish disease heritage genes[2,9]. Violin plots are scaled to have the same area. Box plots within violins show the 1st, 2nd and 3rd quartiles of the MAF distribution; whiskers maximally extend to 1.5 interquartile range.

**a**

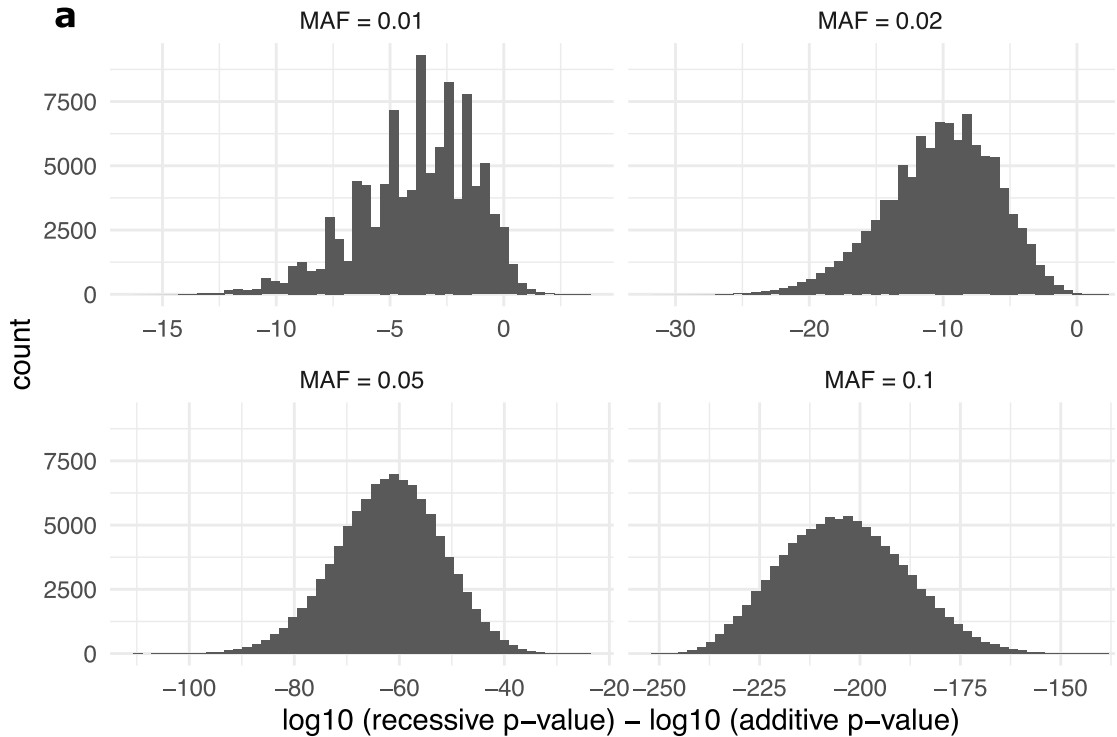

log10 (recessive p–value) − log10 (additive p–value)

**b**

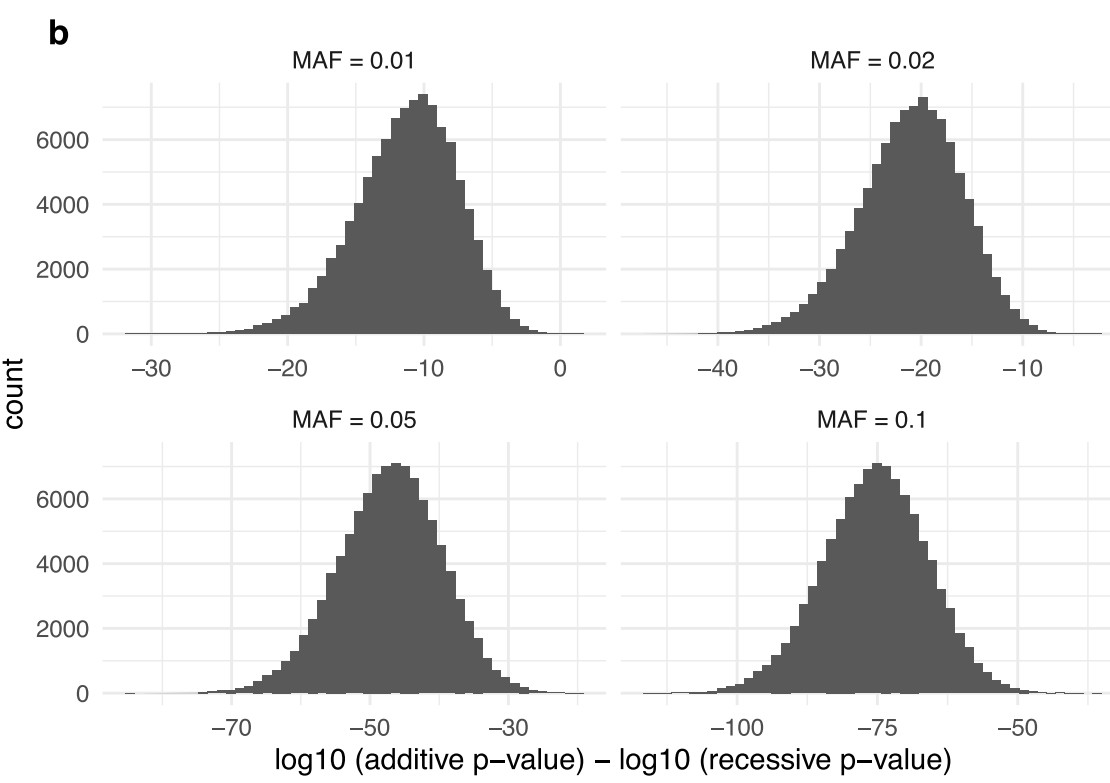

log10 (additive p–value) − log10 (recessive p–value)

**Extended Data Fig. 2 | Simulations of recessive and additive effects at different MAFs.** Here, we generated genotype counts of wild types, heterozygotes and mutant homozygotes in 200,000 individuals of a variant with an allele frequency of 0.01 following Hardy Weinberg Equilibrium (R library: HardyWeinberg) and random controls and cases of a disease with a prevalence of 0.05 (see also Supplementary Note 2). In **a** we simulate a recessive association. Here, we set the probability of homozygotes to develop the disease to 5x compared to wild type and the heterozygous effect to 1 (= no effect). In this histogram we show on the x-axis, the log10 p-value of the recessive model - the log10 p-value of the additive model (method: logistic regression). In **b** we simulate an additive association. Here, we set the homozygous effect to 1.5x and the heterozygous effect to 2.25. In this histogram we show on the x-axis, the log10 p-value of the additive model - the log10 p-value of the recessive model (method: logistic regression).

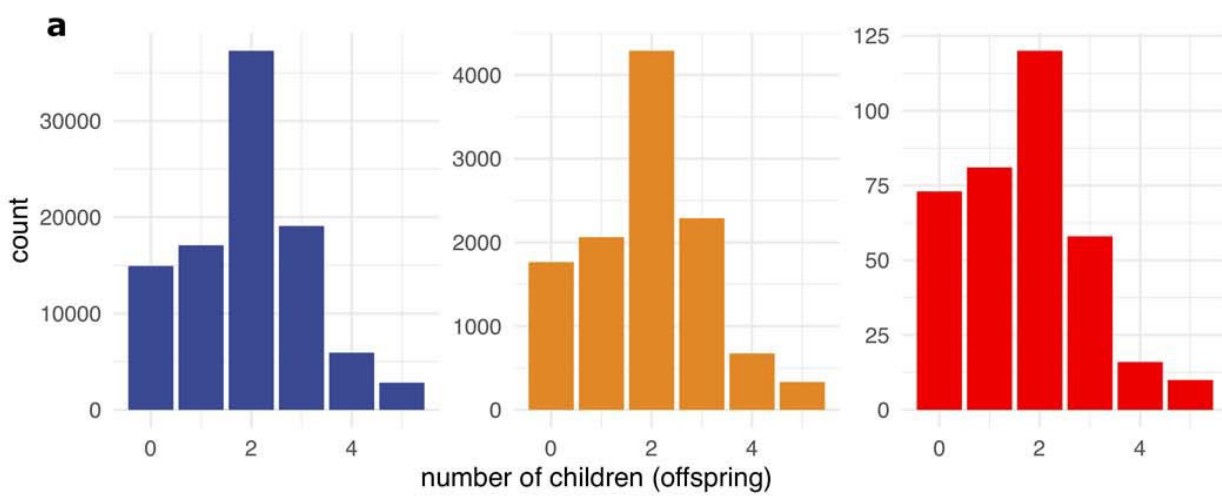

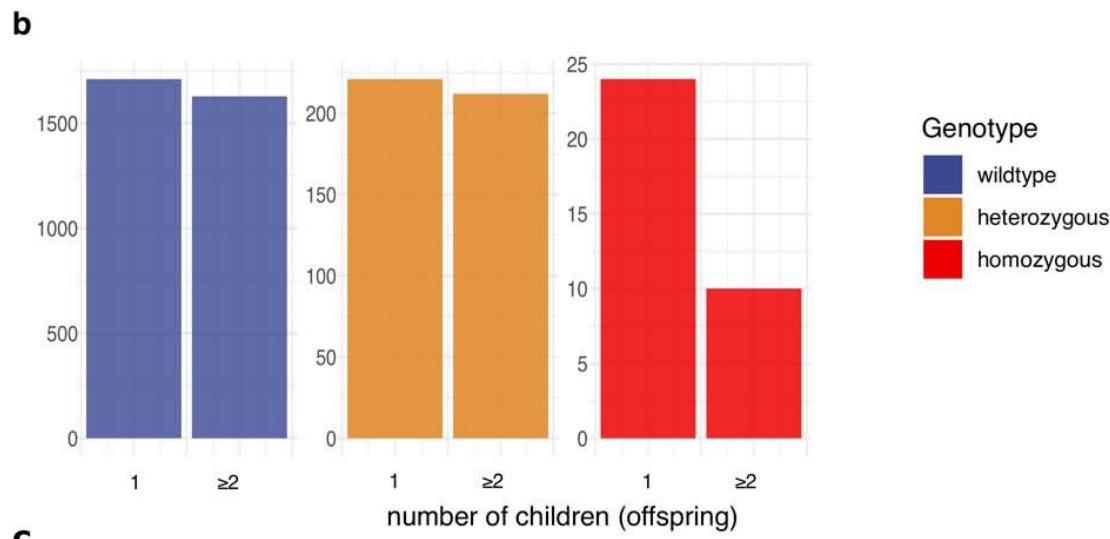

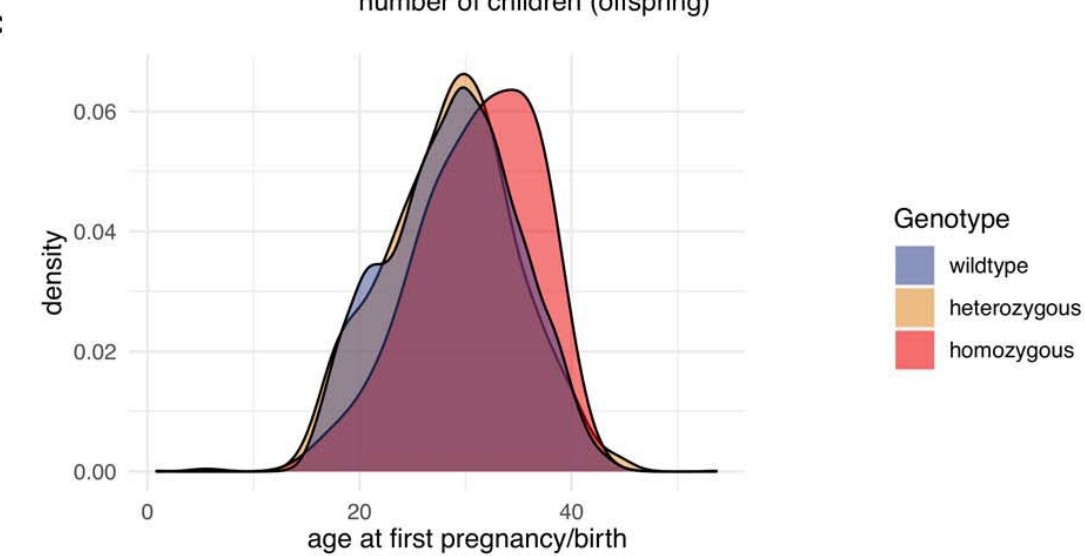

**Extended Data Fig. 3 | *EBAG* is associated with female infertility.**
**a**, Comparing number of offspring in 147,061 women who are wild type
(*n* = 131,141), heterozygous (*n* = 15,455) or homozygous (*n* = 465) for the *EBAG9*
variant. Among 7,980 women with children and diagnosed with infertility 71
*EBAG9* homozygotes (**b**) had fewer children and (**c**) had their first child
significantly later (see Supplementary Note 5).

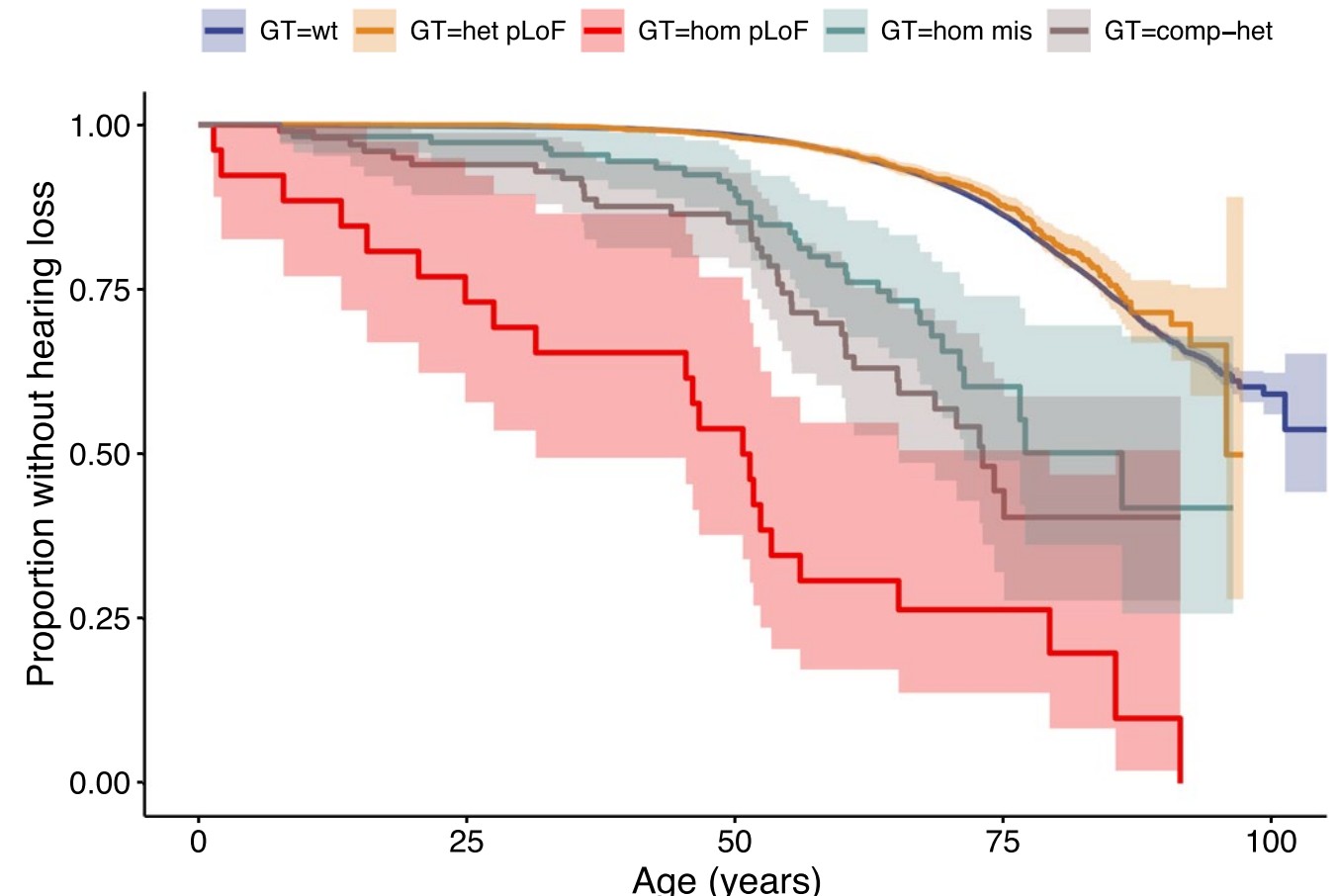

**Extended Data Fig. 4 | Age at first disease diagnosis of variant carriers in *GJB2* (survival plot).** Wt, wild type; het, heterozygous; hom, homozygous; comp-het, compound heterozygous; GT, genotype. Genotypes of a known pathogenic missense and pLoF variant in *GJB2* associated with hearing loss. Comp-het carry both the pathogenic missense and pLoF variant on different alleles.

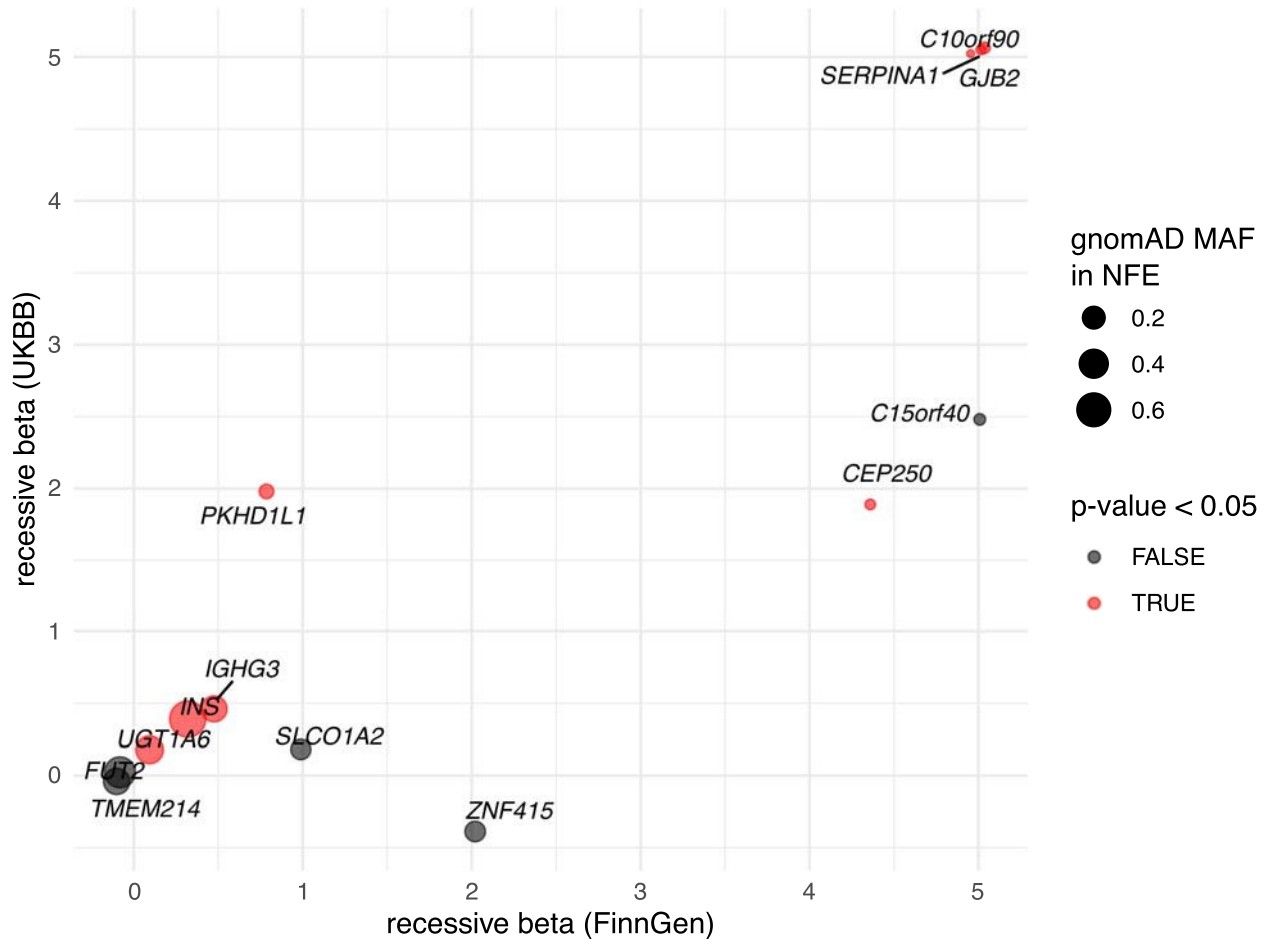

**Extended Data Fig. 5 | Replications with UKBB data.** We replicated the 31 recessive associations in FinnGen in the UKBB[26] with a recessive model in SAiGE[23]. Of the 31 variants, 13 had ≥ 5 homozygotes in the UKBB of which 8 had a significant recessive p-value in UKBB for the same/similar phenotype. Here, we show recessive betas in FinnGen versus UKBB (betas set to 5, when beta > 5). Plot size corresponds to MAF in gnomAD non-Finnish Europeans and plot colour red means the recessive p-value in UKBB is < 0.05.

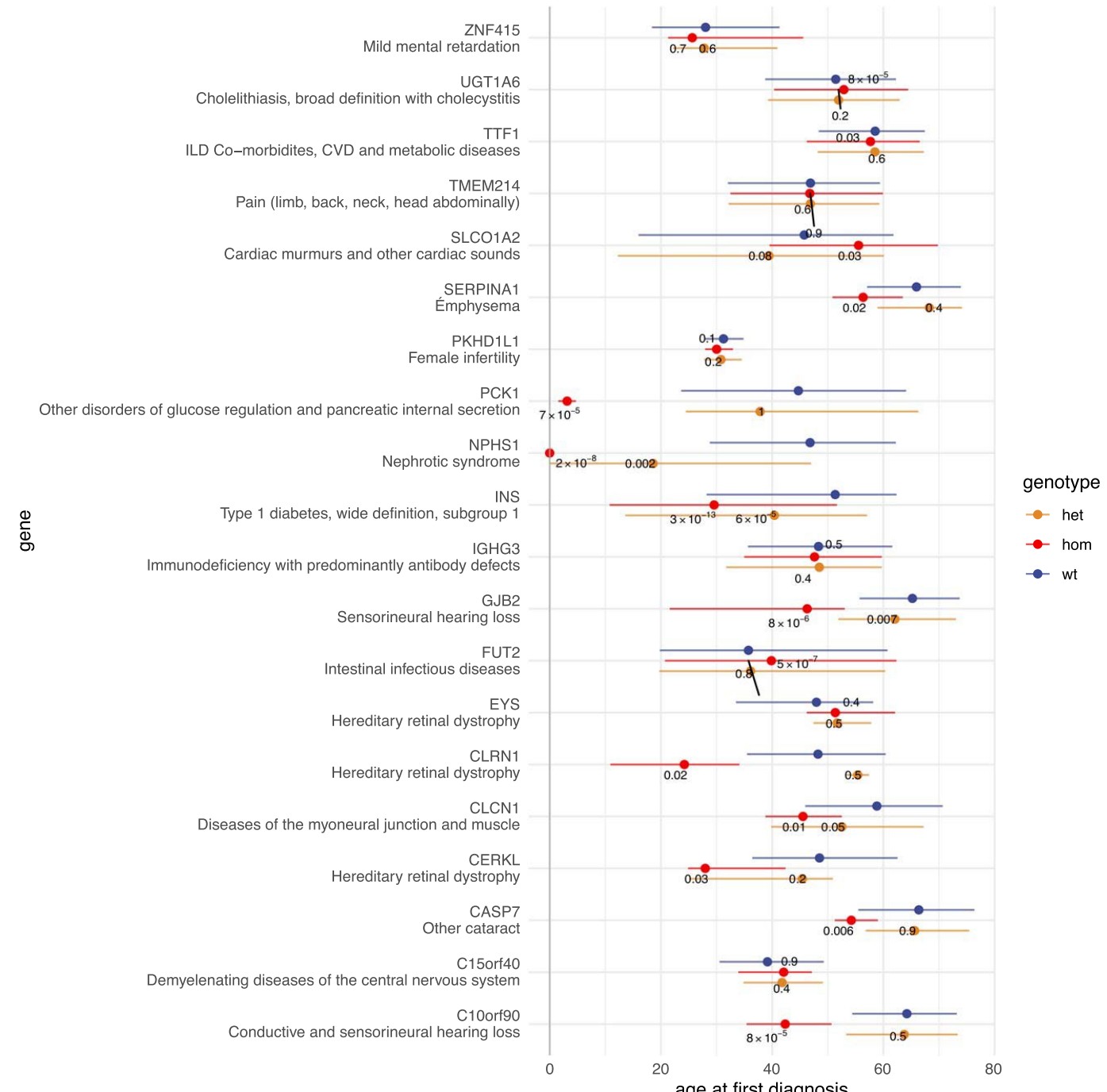

**Extended Data Fig. 6 | Age at disease onset.** This forest plot shows the median age at first diagnosis for each variant with recessive associations in our FinnGen data. P-values indicate differences in disease onset between respective homozygous or heterozygous compared to wild-type carriers (Wilcoxon rank tests). Bars represent the first and third quartile of age at first diagnosis. Only variants with more than 5 affected homozygotes are shown.

The y-axis lists gene-disease associations. Homozygotes had significantly earlier (or later for a known homozygous protective variant in *FUT2*[55]) disease onset for 7/31 variants than wild types (p-value < 0.0016 with Bonferroni correction for 31 tests, 14/31 tests with nominal p-value < 0.05, Wilcoxon rank test.).

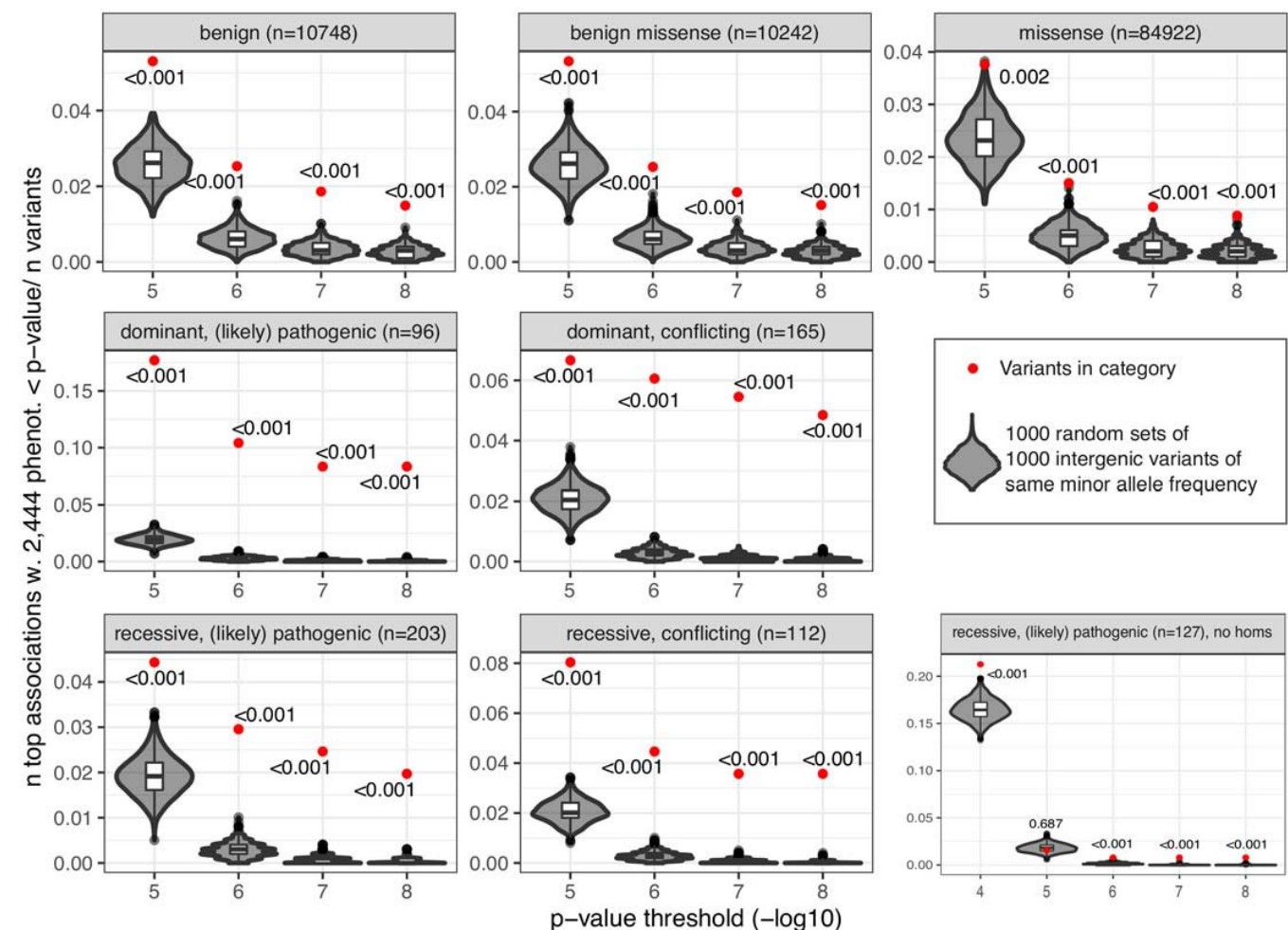

**Extended Data Fig. 7 | Global disease associations of variant categories.**
Variants previously described as disease-causing (ClinVar likely pathogenic or conflicting variants[20]) but also ClinVar likely benign variants are globally associated with disease phenotypes compared with randomly sampled intergenic variants matched to the same minor allele frequency in 15 bins. Disease-causing variants in OMIM genes that were described with only dominant inheritance, as well as genes with only recessive inheritance were globally disease-associated.

# Reporting Summary

Nature Research wishes to improve the reproducibility of the work that we publish. This form provides structure for consistency and transparency in reporting. For further information on Nature Research policies, see our Editorial Policies and the Editorial Policy Checklist.

## Statistics

For all statistical analyses, confirm that the following items are present in the figure legend, table legend, main text, or Methods section.

| n/a | Confirmed | |
|---|---|---|
| ☐ | ☒ | The exact sample size (*n*) for each experimental group/condition, given as a discrete number and unit of measurement |
| ☐ | ☒ | A statement on whether measurements were taken from distinct samples or whether the same sample was measured repeatedly |
| ☐ | ☒ | The statistical test(s) used AND whether they are one- or two-sided<br>*Only common tests should be described solely by name; describe more complex techniques in the Methods section.* |
| ☐ | ☒ | A description of all covariates tested |
| ☐ | ☒ | A description of any assumptions or corrections, such as tests of normality and adjustment for multiple comparisons |
| ☐ | ☒ | A full description of the statistical parameters including central tendency (e.g. means) or other basic estimates (e.g. regression coefficient) AND variation (e.g. standard deviation) or associated estimates of uncertainty (e.g. confidence intervals) |
| ☐ | ☒ | For null hypothesis testing, the test statistic (e.g. *F*, *t*, *r*) with confidence intervals, effect sizes, degrees of freedom and *P* value noted<br>*Give P values as exact values whenever suitable.* |
| ☐ | ☒ | For Bayesian analysis, information on the choice of priors and Markov chain Monte Carlo settings |
| ☐ | ☒ | For hierarchical and complex designs, identification of the appropriate level for tests and full reporting of outcomes |
| ☐ | ☒ | Estimates of effect sizes (e.g. Cohen's *d*, Pearson's *r*), indicating how they were calculated |

*Our web collection on statistics for biologists contains articles on many of the points above.*

## Software and code

Policy information about availability of computer code

| Data collection | Genotype calls were made with GenCall and zCall algorithms for Illumina and AxiomGT1 algorithm for Affymetrix data. Genotype imputation was done with Beagle 4.1 (version 08Jun17.d8b) with the population-specific SISu v3 reference panel created from high-quality WGS data of 3,775 individuals. Further processing of the genotype data was done using standard genome analysis software Plink 1.9 and 2.0 and BCFtools 1.7 and 1.9. |
|---|---|
| Data analysis | We used the SAIGE (version 0.35.8.8) software for running all R4 GWAS with additive and recessive models. Pipelines for parallel computing were created using Cromwell-29 and 31 and Wdltool-0.14. Fine-mapping was done with SuSiE. Further processing of the data was done using R 3.4.1 (packages: data.table 1.10.4, sm 2.2-5.4). Statistical analyses and figures were done using additional R packages ggplot2, plyr, survminer, survival, tidyr, Rutils. |

For manuscripts utilizing custom algorithms or software that are central to the research but not yet described in published literature, software must be made available to editors and reviewers. We strongly encourage code deposition in a community repository (e.g. GitHub). See the Nature Research guidelines for submitting code & software for further information.

## Data

Policy information about availability of data

All manuscripts must include a data availability statement. This statement should provide the following information, where applicable:

- Accession codes, unique identifiers, or web links for publicly available datasets
- A list of figures that have associated raw data
- A description of any restrictions on data availability

All summary statistics described in this manuscript can be found in the Supplementary Tables. A full list of FinnGen endpoints for release 4 is available at https://www.finngen.fi/en/researchers/clinical-endpoints.

# Field-specific reporting

Please select the one below that is the best fit for your research. If you are not sure, read the appropriate sections before making your selection.

☒ Life sciences ☐ Behavioural & social sciences ☐ Ecological, evolutionary & environmental sciences

For a reference copy of the document with all sections, see nature.com/documents/nr-reporting-summary-flat.pdf

# Life sciences study design

All studies must disclose on these points even when the disclosure is negative.

| | |
|---|---|
| Sample size | All 176,899 participants in FinnGen datafreeze 4 were included in the study. |
| Data exclusions | In FinnGen datafreeze 4, individuals with missing minimum phenotype data or a mismatch between imputed sex and sex in registry data as well as population outliers in PCA have been excluded. |
| Replication | All homozygous associations we present in Table 1 including the novel hits we highlight in the manuscript and in a Supplementary Note were validated in FinnGen datafreeze 6 (n=234,553) and/or in the UK biobank (n=420,531). |
| Randomization | GWAS were performed in individuals with versus without one of 2,444 disease endpoints. In all GWAS age, sex, genotyping batch and the first ten PCs of genotypes were included as covariates. |
| Blinding | In hypothesis free GWAS, blinding is not possible/necessary. |

# Reporting for specific materials, systems and methods

We require information from authors about some types of materials, experimental systems and methods used in many studies. Here, indicate whether each material, system or method listed is relevant to your study. If you are not sure if a list item applies to your research, read the appropriate section before selecting a response.

## Materials & experimental systems

| n/a | Involved in the study |
|---|---|
| ☒ | ☐ Antibodies |
| ☒ | ☐ Eukaryotic cell lines |
| ☒ | ☐ Palaeontology and archaeology |
| ☒ | ☐ Animals and other organisms |
| ☐ | ☒ Human research participants |
| ☐ | ☒ Clinical data |
| ☒ | ☐ Dual use research of concern |

## Methods

| n/a | Involved in the study |
|---|---|
| ☒ | ☐ ChIP-seq |
| ☒ | ☐ Flow cytometry |
| ☒ | ☐ MRI-based neuroimaging |

## Human research participants

Policy information about studies involving human research participants

| | |
|---|---|
| Population characteristics | Participants are on average 59.4 ± 17 (mean ± standard deviation) years old, 100,361 are female, 76,538 male. As samples were mainly collected through legacy collections and hospital biobanks the estimated FinnGen participant may be expected to have more diseases than Finnish population average. Phenotype data comes from digital health record data from Finnish health registries. |
| Recruitment | The collected samples consist of two entities: 1) legacy samples, mainly collected by the THL (National Institute for Health and Welfare in Finland) and 2) prospective samples which were mainly be collected by hospital biobanks. Almost all of the Finnish biobanks are part of the FinnGen study. |
| Ethics oversight | The FinnGen project has been approved by the Coordinating Ethics Committee of the Helsinki and Uusimaa Hospital District. |

Note that full information on the approval of the study protocol must also be provided in the manuscript.

## Clinical data

Policy information about clinical studies

All manuscripts should comply with the ICMJE guidelines for publication of clinical research and a completed CONSORT checklist must be included with all submissions.

| | |
|---|---|
| Clinical trial registration | *Provide the trial registration number from ClinicalTrials.gov or an equivalent agency.* |

Study protocol    *Note where the full trial protocol can be accessed OR if not available, explain why.*

Data collection    *Describe the settings and locales of data collection, noting the time periods of recruitment and data collection.*

Outcomes    *Describe how you pre-defined primary and secondary outcome measures and how you assessed these measures.*

