## [Peer Review File · Nature]

Manuscript Title: Mono- and biallelic variant effects on disease at biobank-scale.

Reviewer Comments & Author Rebuttals

Reviewer Reports on the Initial Version:

Referees' comments:

Referee #1 (Remarks to the Author):

Review of "Recessive Mendelian effects of coding variants on disease in 176,899 Finns" by Heyne et al

This paper sets out to use a combined EHR and sequencing approach for a broad-based assessment of the effect of rare, homozygous variants on the phenotype of individuals in Finland. Major Points

1. There is a serious terminology problem with this manuscript. I suspect that this is because these authors are translating a GWAS world view onto Mendelian traits. One aspect is inheritance patterns. In the GWAS world, it is acceptable to say "recessive effects" because of the additive nature of the alleles in the association tests – but since they are complex traits, that never gets translated to inheritance patterns. Here, they are jumping from these additive association patterns into the world of Mendelian or single gene inheritance, so the terminology has to work in both. The authors could well-serve the readership and the field if they were to recognize that some of the effects they have observed would be better described as semi-dominant (incompletely dominant is an acceptable synonym). Just because they have identified an association for the biallelic variant state does not mean that the inheritance pattern will be autosomal recessive – it could indeed be semi-dominant. They could also say "Novel biallelic GWAS associations...", but since they are making much of the homozygosity in this population, they might want to stick with the former. If they have solid data that there are no detectable phenotypic consequences of the heterozygous state, then designating it as AR would be fine. In fact, I would think this data set would be very powerful for doing that. If there are potential heterozygote phenotypes, then they should instead designate it as semidominant. I think that when they say "with additive and recessive models" they mean semidominant and autosomal recessive. I wonder if, when they say "dominant/recessive" they mean semidominant? It would help enormously to untangle this confusion. I also wonder if that is what they mean when they say "dominant/additive". Once I got to the discussion, I could see that much of this is a paper tiger. Throughout the paper, they misuse genetic terminology and then in the discussion show us that they know this is problematic. I don't get it. Why wouldn't they adopt formal, careful terminology at the outset, use it consistently throughout the paper and spare us this "lesson"? This issue applies to the title as well.
2. The authors should consider that they have an opportunity here to do much more. If I am reading these data correctly, they are providing evidence to support the model that quite a few human Mendelian inheritance patterns are semidominant. This is very important and could allow this paper to have very high impact. By systematically studying datasets such as these, they can begin to build a more sophisticated and nuanced understanding of this. It is always a challenge to build a language and syntax that correctly describes biology. We have had a fairly useful, but ultimately flawed language and syntax for 'mendelian' traits – a strong justification for this paper appearing in Nature could be that they thoughtfully tackle this interesting and important challenge. They broach this with "We therefore argue that semi-dominant inheritance describes the effect of most variants in tumor suppressor genes more adequately than dominant inheritance." – but I think they could go much further as they have identified this effect as well in non-cancer associated variants. I think it is time for a re-evaluation of our prior, simplistic concepts. While Mendel had profound insights, humans are more complex than peas, and more nuanced and biologically valid insights can be made.
3. A related issue is that there is no such thing as a "homozygous carrier". Per PMID 27657676,

the term “carrier” should only be applied to an (asymptomatic) heterozygote for a variant associated with a disorder inherited in an autosomal recessive pattern. An acceptable term would be “homozygotes” or they could say “individuals harboring homozygous variants”. A related issue is this: “With longitudinal EHR data we also find that homozygous carriers had disease diagnoses earlier than wildtype carriers.” I have no idea what a “wildtype carrier” is. If it refers to homozygous wildtype genotype, that is a tautology – of course affecteds have diagnoses earlier than unaffecteds – what are they trying to say? Or instead, perhaps it is an odd way of describing heterozygotes? That would then fit in to the semidominant point raised above. They really need to straighten out their genetic terminology. It is hard to know what they are intending to say. There are three possible genotypes for a biallelic SNP: homozygous wildtype (which can just be called wildtype), homozygous mutant, (which cannot just be called ‘mutant’ because it can be confused with the final term), and heterozygous. All of these uses of “carrier” have to be stripped out of the paper as they muddle things terribly.

4. The authors, understandably, see primarily the merits in their approach. “Rare genetic variants with large effects on disease can have potential direct treatment implications; studying their effects comprehensively requires however large sample sizes.” A weakness of their approach is that the phenotyping is broad, but it is not deep. They clearly state that the phenotype space comprises 2,444 phenotypic endpoints. Do they believe that that comprises all possible phenotypes? Of course it does not and they cannot detect those, irrespective of how many participants they study with this design. In the discussion they claim that these are “high-quality phenotypes” – but there is no external measure of the “quality”, whatever that is purported to mean. That claim must be removed, unless they can demonstrate the external validity of this claim. The weakness of their approach is depth – subtle findings not appreciated by ordinary clinicians (e.g., CSF neurotransmitter levels) would likely never be detected. So, while I appreciate the power of what they are doing and the validity of their findings, they must acknowledge its weaknesses and limitations. Deep, bespoke phenotyping can identify phenotypes invisible to this method. A common problem with all GWAS and similar association work is that a p-value is not a conclusion – it is just a statistical metric. As well, one could ask – what is next? Just because an association has been identified by this approach, does that make it real? Ought they not advocate for performing a genotypic ascertainment approach to actually have clinicians identify individuals with these genotypes and confirm these findings? I would suggest that the word “candidate” ought to be added to their conclusions – they must be considered tentative until replicated. E.g., “In summary, we identify 31 Mendelian recessive disease associations...” I would suggest the word “candidate” should be inserted between “disease” and “associations”. In the GWAS world, a p-value seems to be taken at face value, but in Mendelian genetics, we generally expect replication, allelic series, linkage, etc. to buttress a claim of associating a gene with a disease.

5. Another weakness they do not address is that the large regions of homozygosity can make it difficult to determine which of the many variants in a haplotype block are the actual causative variant. They have successfully overcome this in a number of examples, but it is still an issue and less of a problem when studying outbred populations. This must be acknowledged. What we don’t need is a sales pitch for populations with bottlenecks and little outbreeding. What we do need is a balanced, sober view of the strengths and limitations of the approach.

6. This makes no sense: “These included 311 coding variants that were annotated as (likely) pathogenic, 147 of them by multiple submitters and thus regarded as pathogenic with “high confidence”. If they were annotated as ‘likely pathogenic’ by multiple submitters, how can they be regarded as pathogenic with high confidence? I don’t understand the logic of this.

7. I don’t know what the solution to this could possibly be, but the simplistic notion of calling an interpretation from ClinVar as “conflicting” obscures a host of issues. There are trivial conflicts and major ones. For example, a variant could be labeled “conflicting” just because of an OMIM assertion is foolish – the OMIM variant interpretations are the least reliable in all of ClinVar. Maybe this doesn’t matter to the conclusions of this paper? Some thoughtful discussion of the limitations of this use of ClinVar would be appropriate.

Minor points

1. Formally, neither variants nor diseases nor genes have the attribute of “autosomal recessive”. It

is inheritance patterns that have that attribute. This becomes apparent when one thinks about tumor suppressor gene-related cancer syndromes. These disorders are recessive in a pathogenetic sense, but the inheritance pattern of the cancer susceptibility is autosomal dominant. These concepts are distinct in important ways. Also, "...and show mostly recessive inheritance..." should be amended to "autosomal recessive". These authors apparently forgot that there is an inheritance pattern described as X-linked recessive. What is really remarkable is that in the discussion the authors say exactly this, but violate it repeatedly.

2. In the list of prior studies of bottleneck, inbred populations, it would be better to say "Anabaptist sects" rather than Mennonites, as the Amish and Hutterites have been just as informative.
3. It would be better to keep all language formal. "...long runs of homozygosity..." is common, but slang. "Hits" is slang. "...and thus permitted at higher frequencies..." No one permits these variants – they are observed at higher frequency than is the case for variants for disorders with an abnormal phenotype in heterozygotes. "Brits" is slang.
4. I have to guess that what they mean with the notation "(likely) pathogenic" is it a collapse of these two categories of pathogenicity classification into one? I.e., "pathogenic or likely pathogenic"?
5. Whenever one makes a comparison, it is essential to specify the comparator. It can sometimes be inferred, but it should not have to be. E.g., "Effects of known Mendelian heterozygous variants in our dataset are thus predominantly "milder" including variants in cancer genes like PALB2, CHEK2 and JAK2. Milder than what?"
6. "In the 19th century Mendel coined the term 'recessive inheritance' if a particular phenotype only appeared in organisms that were homozygous for a given allele. That definition is still used in genetics today." No, it is not. One has to widen that definition to include individuals who are compound heterozygotes. Mendel was wrong about this for an understandable reason – he was describing traits in a small population with allelic homogeneity. The abnormal biallelic genotype could only be homozygous.
7. Some of the discussion is fairly pedantic. The point about heterozygote advantage is presented in very unsophisticated manner. Anyone who will read this paper will know what heterozygote selection and advantage is and the discussion should not read like a graduate student textbook. It should instead focus on what is novel in this manuscript and an expansion on the conclusions, further directions, etc.
8. This statement, which is early in the discussion needs some clarification: "Disease-causing variants are enriched to unusually high frequencies in populations with a history of recent bottlenecks such as Finland." Ought this not to be clarified to make clear that this applies to subset of variants? The Finns do not (I think) have an overall higher prevalence of genetic disease than do out bred populations. They do not have unusually high frequencies of disease-associated variants in toto - I think it is the case that they have unusually high frequencies of a few variants, and a deficiency of others. Bottlenecks, I believe, both ratchet up and down.

Referee #2 (Remarks to the Author):

The paper by Heyne et al. summarises the recessive effects of coding variants across numerous diseases in a large cohort from the Finnish population. It combines improved interpretation of known clinical variants previously classified "pathogenic" or "benign", including an evaluation of penetrance and disease associations with heterozygous and recessive models, as well as novel gene discovery.

The paper is excellent. It is clearly written, well referenced, and contains multiple novel and important findings. The data and methodology are robust and the results have been presented clearly. Statistical methods are appropriate and confidence intervals are included on graphs where needed. In my opinion, the conclusions are robust.

I have the following minor comments:

-The section in the discussion on semi-dominant effects, specifically in relation to tumour-suppressor genes, seems to come out of the blue. Given the rest of the paper focuses on recessive diseases, it is unclear why the focus here shifts to tumour suppressor genes. I suggest either adding a Figure (or referring to a supplementary figure) showing data supporting the authors' opinions about CHEK2/JAK2, or limiting their comments relating to semi-dominant effects in these genes.

- Supplementary Figure 4 contains a survival curve for INS variants in type 1 diabetes showing very little effect. However, INS mutations are linked to monogenic diabetes (MODY, TNDM); have the authors tested the penetrance of INS variants in monogenic or very early onset diabetes? This would be more appropriate than type 1.

- The authors mention the fact that the term "recessive" is often misleading when there are clear phenotypic effects in carriers. This is true and quite well known, and a good example to add might be beta-thalassaemia minor. However, there are also examples of truly recessive diseases, such as (according to the data in Supplementary Table 4) nephrotic syndrome caused by recessive variants in NPHS1 – I thought this an interesting finding it might be worth highlighting in the text.

- Table 1. Please clarify if the MAF is Finnish or NFE.

- Table 2. I am unable to see the variant positions referred to in the table legend.

- Please check Figure numbering, I think it has gone awry; there is no Figure 3C and I think Supplementary Figure S5 is not referred to in the text.

Referee #3 (Remarks to the Author):

The authors describe a large recessive analysis in Finnish biobank data. The methods and analyses are sound, and there are lots of results here.

I found it difficult to follow any kind of consistent narrative or explanation of the results. There were a lot of enumerations of numbers of different kinds of observations, but it was hard to see what was new, or how the results together showed a major new understanding of the genetic architecture of these diseases. I give some examples below.

Comments:

Figure 1 shows two striking differences in recessive disease causing variants between Finns and an equal number of non-Finnish Europeans: (a) higher frequency and (b) fewer at non-zero frequency. I think (a) can only be explained by the unique population history in Finland, but (b) seems like it might be affected as well by the fact that the NFE group presumably comes from many countries. What would it look like to compare to 10,000 French, or Dutch, etc.?

What's the interpretation of the paragraph from lines 136 to 151? There's a listing of inconsistencies with ClinVar and OMIM, but I think it's not that surprising to find some of these. And if they are surprising, it needs explaining. In terms of annotated dominant variants in this (apparently) unaffected population, are the authors certain the individuals have no phenotype? And this isn't really a particularly good dataset for answering this question, since the special properties of the Finnish population affect heterozygous frequencies less (as noted above, they bring some things from low frequency to intermediate, but push lots of others down to zero). Then the PALB2 story is explained in the discussion in such a way that basically there's no surprise, so the whole thing is kind of an obtuse critique of OMIM terminology. And then for the recessive genes, some are surprising, like 500 homozygotes at BTBD, but nothing is made of it (maybe biotinidase deficiency isn't that uncommon?) And GJB2 is then featured as a finding in the paper later on.

How do the authors explain the apparent conflict of direction of effect of SCN5A?

I couldn't find recessive stats on r4.finngen.fi?

line 66: " Multiple bottleneck events in the Finnish population history" do these amplify each other (i.e. are they consecutive?)

line 79: perhaps replace "ultrarare" with a number, esp. if there is an estimate of what frequency range the enrichment covers

line 167: what do one star and two star reviews mean in ClinVar? (i.e. which is better, and how high/low does the scale go)

line 176: what are "non Finnish Swedish Estonian Europeans", and why are they a control group?

line 189: "similar to known recessive" <- what were they?

lines 258-261: was a bit confused that there are four variants, two in each, and in each case to gene-consistent phenotype (I think). perhaps unpack a bit to make clearer.

line 308 and elsewhere: "homozygous carriers" is not a common phrase, and can be confusing with "carrier" usually meaning heterozygous for an allele that causes disease when homozygous. homozygote genotypes, or just homozygotes? (unless you mean something else)

line 475: "Brits with Pakistani ancestry" is pretty casual

line 648 and elsewhere: "please see the flagship paper" <- what's this?

Author Rebuttals to Initial Comments:

Referees' comments:

Referee #1 (Remarks to the Author):

Review of “Recessive Mendelian effects of coding variants on disease in 176,899 Finns” by Heyne et al

This paper sets out to use a combined EHR and sequencing approach for a broad-based assessment of the effect of rare, homozygous variants on the phenotype of individuals in Finland.

Major Points

1. There is a serious terminology problem with this manuscript. I suspect that this is because these authors are translating a GWAS world view onto Mendelian traits. One aspect is inheritance patterns. In the GWAS world, it is acceptable to say “recessive effects” because of the additive nature of the alleles in the association tests – but since they are complex traits, that never gets translated to inheritance patterns. Here, they are jumping from these additive association patterns into the world of Mendelian or single gene inheritance, so the terminology has to work in both. The authors could well-serve the readership and the field if they were to recognize that some of the effects they have observed would be better described as semi-dominant (incompletely dominant is an acceptable synonym). Just because they have identified an association for the biallelic variant state does not mean that the inheritance pattern will be autosomal recessive – it could indeed be semi-dominant. They could also say “Novel biallelic GWAS associations...”, but since they are making much of the homozygosity in this population, they might want to stick with the former. If they have solid data that there are no detectable phenotypic consequences of the heterozygous state, then designating it as AR would be fine. In fact, I would think this data set would be very powerful for doing that.

If there are potential heterozygote phenotypes, then they should instead designate it as semidominant. I think that when they say “with additive and recessive models” they mean semidominant and autosomal recessive. I wonder if, when they say “dominant/recessive” they mean semidominant? It would help enormously to untangle this confusion. I also wonder if that is what they mean when they say “dominant/additive”. Once I got to the discussion, I could see that much of this is a paper tiger. Throughout the paper, they misuse genetic terminology and then in the discussion show us that they know this is problematic. I don’t get it. Why wouldn’t they adopt formal, careful terminology at the outset, use it consistently throughout the paper and spare us this “lesson”? This issue applies to the title as well.

This is a very good point and one we have put a considerable amount of effort into in the revision. As one aim of our manuscript is to speak to both genetics researchers from the GWAS as well as Mendelian communities we aimed to find a consistent nomenclature that is acceptable in both groups.

In the quantitative genetics literature (a nice explication in Lander & Botstein, Genetics, 1989) additive (a, representing effect of variable with values 0, 1, 2 according to genotype) and dominant (d, [0, 1, 1]) refer to two independent terms in the 2 degree of freedom (d.f.) linear model. In clinical and rare disease genetics on the other hand - dominant and recessive have strict and well understood meanings. So potential confusion arises by ‘dominant’

referring to the model of Mendelian inheritance while also to the 'dominance' term of the association model in quantitative genetics. For clarity, we now state our definitions of inheritance patterns and how they relate to mono- and biallelic effect sizes in a schematic Figure 2.

In the revised manuscript we modified the terminology again with the reviewers' suggestions in mind, more clearly separating terms referring to inheritance and variant effect, respectively. We found this somewhat difficult for variants with both heterozygous and homozygous effects but quite disproportionately larger effect size in homozygous than in heterozygous state exceeding a linear additive model (termed non-additive effects in quantitative genetics). After extensive discussion with Dr. Heidi Rehm, we decided to term the inheritance of those variant effects 'recessive with variable expressivity in heterozygotes' which we find suitable and practical (see Figure 2).

For our homozygous associations we had indeed repeated the disease association tests after excluding homozygotes to test for the presence of heterozygous effects and now show those results more prominently in Table 2. We find small nominally significant heterozygous effects for some of the variants with large homozygous effects (including for variants with known recessive inheritance such as a pLoF in *GJB2* associated with hearing loss).

2. The authors should consider that they have an opportunity here to do much more. If I am reading these data correctly, they are providing evidence to support the model that quite a few human Mendelian inheritance patterns are semidominant. This is very important and could allow this paper to have very high impact. By systematically studying datasets such as these, they can begin to build a more sophisticated and nuanced understanding of this. It is always a challenge to build a language and syntax that correctly describes biology. We have had a fairly useful, but ultimately flawed language and syntax for 'mendelian' traits – a strong justification for this paper appearing in Nature could be that they thoughtfully tackle this interesting and important challenge. They broach this with "We therefore argue that semi-dominant inheritance describes the effect of most variants in tumor suppressor genes more adequately than dominant inheritance." – but I think they could go much further as they have identified this effect as well in non-cancer associated variants. I think it is time for a re-evaluation of our prior, simplistic concepts. While Mendel had profound insights, humans are more complex than peas, and more nuanced and biologically valid insights can be made.

While we are best-powered in cancer and identify multiple examples where semi-dominant inheritance clearly more adequately describes inheritance than labels of 'recessive' and 'dominant' we indeed also find evidence of more complex inheritance patterns in non-cancer phenotypes. We communicate this now more clearly in the abstract and discussion (ll. 535 ff.). We also performed following additional analyses to bolster those findings:

1) We repeated our global burden test with variants that were previously associated only with recessive inheritance in OMIM after excluding homozygous carriers. We still found a global phenotype association signal that cannot be driven by homozygous associations and therefore most likely comes from heterozygous effects.

2) We performed simulation analyses (Supplementary Note, Supplementary Figure 4) showing that an additive model catches large homozygous effects

poorly in the presence of no heterozygous effect and increasingly better with increasing effect sizes of heterozygous effects. This supports the presence of heterozygous effects in our global association signal of disease variants with recessive inheritance.

3. A related issue is that there is no such thing as a “homozygous carrier”. Per PMID 27657676, the term “carrier” should only be applied to an (asymptomatic) heterozygote for a variant associated with a disorder inherited in an autosomal recessive pattern. An acceptable term would be “homozygotes” or they could say “individuals harboring homozygous variants”. A related issue is this: “With longitudinal EHR data we also find that homozygous carriers had disease diagnoses earlier than wildtype carriers.” I have no idea what a “wildtype carrier” is. If it refers to homozygous wildtype genotype, that is a tautology –

The proposed terminological changes have been made.

of course affecteds have diagnoses earlier than unaffecteds – what are they trying to say?

We want to say that in our data affecteds carrying a genetic variant have diagnoses earlier than affecteds NOT carrying the particular genetic variant. We think, this is additional evidence, that the genetic variant has a true effect on disease. We revised the section in the manuscript (ll. 357 ff.).

Or instead, perhaps it is an odd way of describing heterozygotes? That would then fit in to the semidominant point raised above. They really need to straighten out their genetic terminology. It is hard to know what they are intending to say. There are three possible genotypes for a biallelic SNP: homozygous wildtype (which can just be called wildtype), homozygous mutant, (which cannot just be called ‘mutant’ because it can be confused with the final term), and heterozygous. All of these uses of “carrier” have to be stripped out of the paper as they muddle things terribly.

1) The terminology describing genotypes as been adapted as suggested.

2) All instances of the use of “carrier” as described above were removed from the manuscript.

4. The authors, understandably, see primarily the merits in their approach. “Rare genetic variants with large effects on disease can have potential direct treatment implications; studying their effects comprehensively requires however large sample sizes.” A weakness of their approach is that the phenotyping is broad, but it is not deep. They clearly state that the phenotype space comprises 2,444 phenotypic endpoints. Do they believe that that comprises all possible phenotypes? Of course it does not and they cannot detect those, irrespective of how many participants they study with this design. In the discussion they claim that these are “high-quality phenotypes” – but there is no external measure of the “quality”, whatever that is purported to mean. That claim must be removed, unless they can demonstrate the external validity of this claim.

The term “high-quality phenotypes” was changed to emphasize rather the breadth and longitudinality (rather than depth) of clinical phenotypes. We also elaborate in the new Supplementary Note 1 that we can identify a majority of frequent variants’ known pathogenic effects in multiple different phenotypes serving as a type of positive control.

The weakness of their approach is depth – subtle findings not appreciated by ordinary clinicians (e.g., CSF neurotransmitter levels) would likely never be detected. So, while I appreciate the power of what they are doing and the validity of their findings, they must acknowledge its weaknesses and limitations. Deep, bespoke phenotyping can identify phenotypes invisible to this method.

We are now discussing the limitation of phenotype depth in the discussion (ll. 555 ff.).

A common problem with all GWAS and similar association work is that a p-value is not a conclusion – it is just a statistical metric. As well, one could ask – what is next? Just because an association has been identified by this approach, does that make it real?

We have validated many associations including those that we highlight in the manuscript in Figure 4 in an independent cohort, the UK biobank.

In addition, we now validate all findings in the FinnGen release R6 which we include in Table 2. Secondly, we believe that the longitudinal visualisation and analysis of our data (Supplementary Figures 5) is an additional validation.

Ought they not advocate for performing a genotypic ascertainment approach to actually have clinicians identify individuals with these genotypes and confirm these findings? I would suggest that the word “candidate” ought to be added to their conclusions – they must be considered tentative until replicated.

E.g., “In summary, we identify 31 Mendelian recessive disease associations...” I would suggest the word “candidate” should be inserted between “disease” and “associations”. In the GWAS world, a p-value seems to be taken at face value, but in Mendelian genetics, we generally expect replication, allelic series, linkage, etc. to buttress a claim of associating a gene with a disease.

We partially disagree with that statement as we do replicate the associations that we highlight in the manuscript in the UK biobank. We make this now more prominent in the manuscript. In addition, we now performed a replication with FinnGen data release 6 (n=234,553) as stated above. We now only present the associations we can replicate in Table 2 in the main text and mention the ones we cannot replicate as candidates in the Supplementary Table S2. We agree with the limitations regarding phenotype depth and rephrase this in the discussion as stated above (ll. 555 ff.).

We have also further characterized the association of the *EBAG9* variant with female infertility, as we now show that homozygotes also have fewer children and a later age at their first child than wildtypes in Supplementary Figure S6.

5. Another weakness they do not address is that the large regions of homozygosity can make it difficult to determine which of the many variants in a haplotype block are the actual causative variant. They have successfully overcome this in a number of examples, but it is still an issue and less of a problem when studying outbred populations. This must be acknowledged. What we don't need is a sales pitch for populations with bottlenecks and little outbreeding. What we do need is a balanced, sober view of the strengths and limitations of the approach.

We added limitations in the introduction (ll. 95 ff.) and also in the discussion part of the manuscript (see also other comments).

6. This makes no sense: “These included 311 coding variants that were annotated as (likely) pathogenic, 147 of them by multiple submitters and thus regarded as pathogenic with “high confidence”. If they were annotated as ‘likely pathogenic’ by multiple submitters, how can they be regarded as pathogenic with high confidence? I don’t understand the logic of this.

We meant that we have more trust in the annotation when multiple independent ClinVar submitters agree (2 out of 4 possible stars in ClinVar). We reworded this to make it clearer.

7. I don’t know what the solution to this could possibly be, but the simplistic notion of calling an interpretation from ClinVar as “conflicting” obscures a host of issues. There are trivial conflicts and major ones. For example, a variant could be labeled “conflicting” just because of an OMIM assertion is foolish – the OMIM variant interpretations are the least reliable in all of ClinVar. Maybe this doesn’t matter to the conclusions of this paper? Some thoughtful discussion of the limitations of this use of ClinVar would be appropriate.

We have extracted from the subset of variants with “conflicting interpretations” in ClinVar (n.b., this is a ClinVar designation, not our assessment that they are conflicting) those that have at least one submitter labelling the variant as ‘pathogenic’ or ‘(likely) pathogenic’. We hypothesized, that this group of variants should contain some truly pathogenic variants and therefore be globally disease-associated (despite being noisier than the [likely] pathogenic group). But to avoid confusion we have now removed this class from the main text while the analysis remains as an additional validation in the Supplementary Figure S2.

Minor points

1. Formally, neither variants nor diseases nor genes have the attribute of “autosomal recessive”. It is inheritance patterns that have that attribute. This becomes apparent when one thinks about tumor suppressor gene-related cancer syndromes. These disorders are recessive in a pathogenetic sense, but the inheritance pattern of the cancer susceptibility is autosomal dominant. These concepts are distinct in important ways. Also, “...and show mostly recessive inheritance...” should be amended to “autosomal recessive”. These authors apparently forgot that there is an inheritance pattern described as X-linked recessive. What is really remarkable is that in the discussion the authors say exactly this, but violate it repeatedly.

We previously defined in the methods that for clarity recessive OMIM gene is referring to ‘gene with recessive inheritance in OMIM’ but we agree that it may be better to spell it out and have eliminated the term ‘recessive OMIM gene’ from the manuscript.

2. In the list of prior studies of bottleneck, inbred populations, it would be better to say “Anabaptist sects” rather than Mennonites, as the Amish and Hutterites have been just as informative.

We corrected this.

3. It would be better to keep all language formal. “...long runs of homozygosity...” is common, but slang. “Hits” is slang. “...and thus permitted

at higher frequencies...” No one permits these variants – they are observed at higher frequency than is the case for variants for disorders with an abnormal phenotype in heterozygotes. “Brits” is slang.

We fixed these issues.

4. I have to guess that what they mean with the notation “(likely) pathogenic” is it a collapse of these two categories of pathogenicity classification into one? I.e., “pathogenic or likely pathogenic”?

This was previously only explained in the Methods section, but we now define it in the results part, too.

5. Whenever one makes a comparison, it is essential to specify the comparator. It can sometimes be inferred, but it should not have to be. E.g., “Effects of known Mendelian heterozygous variants in our dataset are thus predominantly “milder“ including variants in cancer genes like PALB2, CHEK2 and JAK2. Milder than what?

We now removed that sentence from the manuscript.

For clarification, by ‘milder effects’ we meant variants with large (i.e. Mendelian) heterozygous disease effects in our data are not expected to cause severe childhood onset disease with large negative effects on reproductive fitness, as their population frequencies are rare but not ultra-rare or de novo.

6. “In the 19th century Mendel coined the term ‘recessive inheritance’ if a particular phenotype only appeared in organisms that were homozygous for a given allele. That definition is still used in genetics today.” No, it is not. One has to widen that definition to include individuals who are compound heterozygotes. Mendel was wrong about this for an understandable reason – he was describing traits in a small population with allelic homogeneity. The abnormal biallelic genotype could only be homozygous.

The part about Mendel has been excluded.

7. Some of the discussion is fairly pedantic. The point about heterozygote advantage is presented in very unsophisticated manner. Anyone who will read this paper will know what heterozygote selection and advantage is and the discussion should not read like a graduate student textbook. It should instead focus on what is novel in this manuscript and an expansion on the conclusions, further directions, etc.

We aimed to explain some concepts more broadly, but we eliminated these parts of the discussion as most readers may indeed know this.

8. This statement, which is early in the discussion needs some clarification: “Disease-causing variants are enriched to unusually high frequencies in populations with a history of recent bottlenecks such as Finland.” Ought this not to be clarified to make clear that this applies to subset of variants? The Finns do not (I think) have an overall higher prevalence of genetic disease than do out bred populations. They do not have unusually high frequencies of disease-associated variants in toto - I think it is the case that they have unusually high frequencies of a few variants, and a deficiency of others. Bottlenecks, I believe, both ratchet up and down.

Yes, this is already briefly mentioned in the introduction and results, but we now slightly expanded those sections (e.g. ll. 95 ff.) and mentioned it also in the discussion.

Referee #2 (Remarks to the Author):

The paper by Heyne et al. summarises the recessive effects of coding variants across numerous diseases in a large cohort from the Finnish population. It combines improved interpretation of known clinical variants previously classified "pathogenic" or "benign", including an evaluation of penetrance and disease associations with heterozygous and recessive models, as well as novel gene discovery.

The paper is excellent. It is clearly written, well referenced, and contains multiple novel and important findings. The data and methodology are robust and the results have been presented clearly. Statistical methods are appropriate and confidence intervals are included on graphs where needed. In my opinion, the conclusions are robust.

I have the following minor comments:

-The section in the discussion on semi-dominant effects, specifically in relation to tumour-suppressor genes, seems to come out of the blue. Given the rest of the paper focuses on recessive diseases, it is unclear why the focus here shifts to tumour suppressor genes. I suggest either adding a Figure (or referring to a supplementary figure) showing data supporting the authors' opinions about CHEK2/JAK2, or limiting their comments relating to semi-dominant effects in these genes.

We have added references to the Supplementary Figures and tried to embed the section about tumor suppressor genes more seamlessly into the main narrative investigating inheritance of Mendelian variants more generally. We also shrank this section in the discussion.

- Supplementary Figure 4 contains a survival curve for *INS* variants in type 1 diabetes showing very little effect. However, *INS* mutations are linked to monogenic diabetes (MODY, TNDM); have the authors tested the penetrance of *INS* variants in monogenic or very early onset diabetes? This would be more appropriate than type 1.

As there exist indeed pathogenic variants in the *INS* gene leading to neonatal diabetes or MODY we searched whether the missense variant rs3842753 could cause a MODY-type of diabetes hiding among the common types. There are unfortunately no specific ICD codes for MODY. The billing code E13 ("Other diabetes") comes closest. The homozygous effect of this variant on E13 (p-value 0.001, beta 0.027) was however not larger than on Type 1 diabetes (p-value 6×10^{-46} , beta 0.028). This did not improve when excluding individuals with a simultaneous diagnosis of T1D and T2D. Interestingly, we additionally found that the variant actually slightly *lowered* risk for type 2 diabetes (p-value 0.0007, beta -0.03) that we could replicate with the additive model in R6 data (additive p-value 2×10^{-9} , beta -0.056).

In the literature, it seems that the association of variants in the *INS* gene with Type 1 diabetes has been known for a while

(<https://www.ncbi.nlm.nih.gov/pmc/articles/PMC1682161/>) and is largely attributed to the 5'VNTR region (<https://www.nature.com/articles/ng1197-350> or). When finemapping the locus for T1D in FinnGen the missense variant is part of the credible set (cs) with a PIP (posterior inclusion probability) of 0.22. The other cs variant (PIP ca. 0.77) is a splice region variant rs689. Both have been described previously (<https://pubmed.ncbi.nlm.nih.gov/25751624/>), rs689 linked to the VNTR

(<https://diabetes.diabetesjournals.org/content/53/7/1884>) and in Finns rs689 is also in LD with our missense variant in question ($r^2=0.9992$). We thus argue that the missense variant most likely tags the known VNTR even if its contribution and the exact mechanism are not completely elucidated.

- The authors mention the fact that the term “recessive” is often misleading when there are clear phenotypic effects in carriers. This is true and quite well known, and a good example to add might be beta-thalassaemia minor. However, there are also examples of truly recessive diseases, such as (according to the data in Supplementary Table 4) nephrotic syndrome caused by recessive variants in NPHS1 – I thought this an interesting finding it might be worth highlighting in the text.

We added beta-thalassaemia as an example for phenotypic effect in carriers. We now highlight the recessive variants in NPHS1 more in the text.

- Table 1. Please clarify if the MAF is Finnish or NFE.

Below the Table we now clarify that the allele frequency refers to MAF in FinnGen.

- Table 2. I am unable to see the variant positions referred to in the table legend.

For reasons of space we excluded the variant positions (and forgot to erase in the legend) as the Table also included rsids, but we now flipped the table 90* and added variant positions back in.

- Please check Figure numbering, I think it has gone awry; there is no Figure 3C and I think Supplementary Figure S5 is not referred to in the text.

We fixed Figure numbering.

Referee #3 (Remarks to the Author):

The authors describe a large recessive analysis in Finnish biobank data. The methods and analyses are sound, and there are lots of results here.

I found it difficult to follow any kind of consistent narrative or explanation of the results. There were a lot of enumerations of numbers of different kinds of observations, but it was hard to see what was new, or how the results together showed a major new understanding of the genetic architecture of these diseases. I give some examples below.

Thanks for this on-target comment - we have made a considerable effort to clarify the message and have rewritten large parts of the paper to now focus better on the main narrative. With biobank-scale additive and homozygous

pheWAS of coding variants we investigated two related questions of interest to the Mendelian and quantitative genetics communities.

1) mono- vs bi-allelic effects of Mendelian disease variants

2) the benefit of homozygous scans in GWAS.

Our results show more complex inheritance patterns and widespread presence of allelic series in Mendelian disease genes which are currently underappreciated in clinical genetics.

Comments:

Figure 1 shows two striking differences in recessive disease causing variants between Finns and a equal number of non-Finnish Europeans: (a) higher frequency and (b) fewer at non-zero frequency. I think (a) can only be explained by the unique population history in Finland, but (b) seems like it might be affected as well by the fact that the NFE group presumably comes from many countries. What would it look like to compare to 10,000 French, or Dutch, etc.?

While we had unfortunately no downsamplings of individual European countries available in gnomAD, we could compare 1712 unique recessive disease causing variants between ca. 13k Swedish (SWE) and ca. 11k Finnish (FIN) individuals, as those neighbouring Nordic populations had roughly similar sample sizes.

There were 570 variants causing recessive disease in the Finns compared to 1504 in the Swedes (in Figure 1 in the manuscript there were 554 variants in 10k Finns vs 2169 variants in 10k non-Finnish Europeans). The ca. 1.2-fold difference in sample size could not explain a three-fold difference in the number of disease causing variants. The corresponding Figure (please see below) looks similar to the figure with NFE combined. We can thus confirm a lower number of variants causing recessive disease in Finns compared to Swedes, which is consistent with an overall reduced rare variant diversity in populations undergoing recent bottlenecks such as Finns (Peltonen, et al, Nat Rev Genet, 2000).

What's the interpretation of the paragraph from lines 136 to 151?

This paragraph is meant to show that we are well-powered to identify known disease associations across a wide range of disease phenotypes in our dataset. We have shortened the paragraph as it does not contain much relevant information towards the main narrative but instead wrote a longer Supplementary Note 1.

There's a listing of inconsistencies with ClinVar and OMIM, but I think it's not that surprising to find some of these. And if they are surprising, it needs explaining.

We now present those findings more concisely. We found most noteworthy that

- 1) benign variants were globally enriched for association and many individually disease-associated with genomewide significance - 16 most likely causal after fine-mapping. We think this is important as benign is often falsely equated with neutral.
- 2) variants in genes with only recessive inheritance were globally disease-associated – we have provided more explanation with further analyses for this particular finding (ll.182 ff.). We are currently submitting variants with novel associations or variants that clarified conflicting or wrong classifications to ClinVar.

In terms of annotated dominant variants in this (apparently) unaffected population, are the authors certain the individuals have no phenotype?

For most high confidence pathogenic variants with dominant inheritance we indeed find disease associations which we now explain in a new Supplementary Note 1. The FinnGen participants are mainly recruited via biobanks in hospitals, so many of them are affected with various diseases which we now mention in the introduction (ll. 104 ff.).

And this isn't really a particularly good dataset for answering this question, since the special properties of the Finnish population affect heterozygous frequencies less (as noted above, they bring some things from low frequency to intermediate, but push lots of others down to zero). Then the PALB2 story is explained in the discussion in such a way that basically there's no surprise, so the whole thing is kind of an obtuse critique of OMIM terminology.

We shrank the section on tumor suppressor genes as it indeed contains much known information. We also included more balanced statements about the advantages and disadvantages of our dataset such as lower numbers of pathogenic variants in different places in the manuscript (e.g. ll. 95 ff.).

And then for the recessive genes, some are surprising, like 500 homozygotes at BTB, but nothing is made of it (maybe biotinidase deficiency isn't that uncommon?) And GJB2 is then featured as a finding in the paper later on. We now provide an explanation for the high number of homozygotes of the variant in the BTB gene in the Supplementary Note 1. This is actually a variant that – in heterozygous state – has been experimentally shown to reduce biotinidase levels by ca. 25%. In homozygous state it should therefore theoretically not be sufficient to cause disease (corresponding to 50% activity equalling a heterozygous null variant). With our lack of homozygous disease association we can thus empirically confirm the experimental hypothesis that the variant should only cause disease when compound heterozygous with a null variant.

How do the authors explain the apparent conflict of direction of effect of SCN5A?

We now provide a mechanistic hypothesis based on published experimental and clinical data for this variant. "Previous experimental data found a mild loss-of-function effect of this variant. This is in line with some ECG times being longer in few SCN5A heterozygotes pointing to a potential slowing of electrical conduction in the heart outlining a possible protective mechanism against cardiac arrhythmia such as atrial fibrillation." (ll. 561 ff.) We are currently starting to investigate this with more ECG data.

I couldn't find recessive stats on r4.finngen.fi?

The paragraph originally referred only to additive stats, but we nevertheless added the recessive summary stats to the public release.

line 66: " Multiple bottleneck events in the Finnish population history" do these amplify each other (i.e. are they consecutive?)

Here 'multiple bottlenecks' refers to consecutive bottleneck events amplifying each other, as well as separate bottleneck events affecting different parts of the Finnish population (Kerminen et al, G3, 2017; Martin et al, AJHG 2018).

line 79: perhaps replace "ultrarare" with a number, esp. if there is an estimate of what frequency range the enrichment covers

We replaced ultrarare with a number: "potentially deleterious (and other) variant classes are specifically enriched at an intermediate frequency range (i.e. ca. 0.5%-5%) in Finland" (Lim et al, Plos Genet 2014; Locke et al Nature 2019).

line 167: what do one star and two star reviews mean in ClinVar? (i.e. which is better, and how high/low does the scale go)

We now provide explanations in ll. 166 ff. In short, for benign variants, ClinVar review quality ranges from zero to three stars, three is the best, the average is 1.2 stars. Thus, poor quality of ClinVar classification cannot explain why so many benign variants in our dataset are disease associated.

line 176: what are "non Finnish Swedish Estonian Europeans", and why are they a control group?

We utilize as a general continental European reference point, exomes from European samples in gnomAD 2.1.1 excluding those from Finland, Sweden and Estonia (so-called, Non-Finnish-Swedish-Estonian Europeans). As there were large-scale migrations from Finland to Sweden in the 20th century, many (on the order of 5%) of the chromosomes from Swedish sequencing studies are of recent Finnish origin, and the linguistically (and geographically) close population of Estonia is likely to share elements of the same ancestral founder effect. This is now also briefly mentioned in the Methods (ll. 739 ff).

line 189: "similar to known recessive" <- what were they?

We now list the known GWAS phenotypes in the manuscript (ll. 208 ff).

They are:

SERPINA1 - Emphysema

XPA – skin cancer

NPHS1 – nephrotic syndrome

lines 258-261: was a bit confused that there are four variants, two in each, and in each case to gene-consistent phenotype (I think). perhaps unpack a bit to make clearer.

We reformulated this to: "Genes *GJB2* and *EYS* each harbored two (likely) pathogenic variants. All of those variants were significantly associated with their expected phenotypes in the recessive model: hearing loss (variants in *GJB2*) or retinal dystrophy (variants in *EYS*)."

line 308 and elsewhere: "homozygous carriers" is not a common phrase, and can be confusing with "carrier" usually meaning heterozygous for an allele that causes disease when homozygous. homozygote genotypes, or just homozygotes? (unless you mean something else)

We corrected this.

line 475: "Brits with Pakistani ancestry" is pretty casual

We corrected this.

line 648 and elsewhere: "please see the flagship paper" <- what's this?

This is a placeholder referring to the FinnGen flagship paper that we aim to publish back-to-back if possible. It is however not absolutely necessary.

Reviewer Reports on the First Revision:

Referees' comments:

Referee #2 (Remarks to the Author):

The authors have substantially rewritten the manuscript taking account of previous comments from the reviewers. I think it is improved and my specific comments from the first version have been addressed. I do have a few additional comments:

- For genes where you detect an effect in heterozygotes, and have excluded homozygotes, how certain are you that the individuals included are not compound heterozygous? Have you tested this, for example by excluding individuals with >2 non-synonymous variants in the gene?

- I'm unclear of the purpose of table 2 and some further discussion might be useful. If the point is merely to say that being "benign" for a condition in ClinVar doesn't mean a variant is neutral for all other conditions, this is obvious; indeed all assertions of pathogenicity must be linked to phenotypes, and variants can only be benign or pathogenic with respect to a phenotype. However, some ClinVar B/LB variants in the table are associated with an increased risk of the same condition. How does this fit with Figure 4? Presumably these are risk alleles that act via a semi-dominant manner but are not causal for monogenic disease? And perhaps it points to the reason that the variant was evaluated by submitters to ClinVar? Also note that at least some of the associations in this table have been seen in previous GWAS.

- Figure 4 is very helpful, but I wonder if it should be figure 1. You refer to the "homozygous model" and the "additive model" early in the manuscript, but these terms come from very different sections of the genomics community, and it was not immediately clear to me what you meant by comparing these terms. Some of the changes in terminology are to respond to previous comments from other reviewers, but to me, additive and homozygous are too different to be compared directly; one is a genotype while the other is a mathematical function. Recessive and semi-dominant are better comparative terms, or you could just use homozygous and heterozygous. For greater consistency with the additive concept, could recessive and dominant be described as threshold models and reduced penetrance and "recessive with rare expressing hets" as multiplicative or exponential models?

Referee #3 (Remarks to the Author):

The authors have revised the paper to address some of the previous comments. I still found it quite hard to follow because of carelessness in how figures are ordered and referenced, unnecessary acronyms, and disordered explanations of unusual terms. All of these can be fixed (I enumerate some below), but they made the review unnecessarily hard going.

There's also a lot of information consigned to supplementary notes. I appreciate the difficulty of putting many results into length limits etc, and the previous review asked for a streamlined main message. But some of them (e.g. supplementary note 6) are mini papers in themselves with results, methods and discussion.

I also have two substantive science comments:

1. On the GWAS analysis, it is interesting that homozygote effects are found that are missed by a standard additive test, but studies from a long time ago, such as the 2007 WTCCC paper, fit multiple models of inheritance in their scans. Of course, in the intervening years the additive model has been such a good fit for most common GWAS associations, that this is no longer widely done. So this is a timely reminder to revisit the topic, but not really a new idea.

That paper (and others) have also used more formal approaches to evaluate the relative fit of recessive vs additive (or 2df) models, which would be better to use than the "2 orders of magnitude p-value" rule of thumb used here.

2. I think the DBH presentation isn't quite right. Its in a section about finding errors in causal variants for monogenic disease. And the variant here is called "probably causal". But probably causal for what? It seems highly unlikely that it is causal for the Mendelian DBH deficiency syndrome, which is what is implied. That syndrome is characterised by complete absence of the DBH enzyme, and this 2011 paper says there are 15 known patients worldwide (<https://www.nature.com/articles/npp201142>). So it seems much more likely that this variant does affect the function of DBH, and thus yields modest lifetime protection from hypertension, but I don't think it should be re-classified as a pathogenic variant in OMIM.

And I suspect this same concept may well play in some of the other cases of benign variants with a detected effect on a related phenotype in the EHRs (e.g. the 16 candidate causal SNPs from supplementary note 6). And isn't this expected from lots of previously published results where GWAS hits are more overlapping than chance with Mendelian genes for related phenotypes? I think it is likely (or certainly possible) that many of these "benign" variants are not neutral, but also that they are not pathogenic in the clinical genetics sense.

Minor:

B/LB and P/LP are unhelpful acronyms. I think it would be better to just use benign and pathogenic, or "likely benign" and "likely pathogenic". I appreciate that the labels are different in OMIM, but for me the cognitive load of parsing them was much higher than remembering that you were including two levels of confidence in the analysis.

Figure 3C is referred to before Figure 2 (or the rest of Figure 3), then Figure 3B is referred to before Figure 3A.

NFSEE is defined the second time it is used, rather than the first

"comp-het" is not a necessary shortening of "compound heterozygous", and is only used once later in the paper anyway

Figure 4 is referenced before all the other figures, because it is framed as a summation in the discussion, but also helpful to refer to earlier, but the order has to be resolved one way or the other.

"FinnGen R6 data" is discussed without really defining what it is. I gather it is an update to the dataset since the initial submission?

Referee #4 (Remarks to the Author):

Heyne and colleagues have utilized Finnish biobank data to examine "Mendelian disease architecture" utilizing the unique population history of the Finnish population to examine bottleneck events that created enrichment for certain variants. They propose to examine homozygous and heterozygous effects of >40,000 coding variants on >2000 disease phenotypes using the EHR data for 176,899 individuals. As a reviewer who did not participate in the first round of review, I am asked to assess whether the authors have adequately responded to a previous review. I am also asked to provide an independent critique of the overall manuscript.

Overall, I like the idea of using "disease architecture" in the title, but I wonder if a more general

phrasing of "architecture of genetic disease" would be more appropriate, given how this work blends GWAS/quantitative genetics with the types of genetic variation that underlies Mendelian/monogenic phenotypes.

One of the main concerns of a previous reviewer was in the use of language to describe inheritance patterns, zygosity, allelic requirement, and disease phenotype. This is in part challenging because of the dual focus on quantitative trait/multifactorial/polygenic disease genetics and Mendelian/monogenic conditions. It can indeed be quite disorienting to toggle between thinking about variants in a "Mendelian" framework where the majority of the phenotype can be attributed to the underlying genotype, versus quantitative genetics in which small polygenic effects contribute as an ensemble. Overall, it appears that the authors have done a good job of addressing this concern, aside from a few instances in which additional clarity may be needed.

On the whole, this manuscript is quite interesting and demonstrates the utility of isolated populations to uncover details of the complexity of genetic influence on disease (which, of course, we have long recognized can cross the traditional divide between polygenic and monogenic).

SPECIFIC COMMENTS:

1. The summary paragraph still includes some slight confusion about the use of term "recessive" which in Mendelian terms should be applied to observed phenotypes (attributable, of course, to biallelic variants). I would simply suggest removing "recessively inherited" from the first sentence. The next sentence would then adequately indicate that the study is interested in homozygous variants that influence recessive diseases, with the slight modification of something like "facilitates the identification of biallelic variants that cause recessive disease phenotypes."

2. The introduction needs to acknowledge the well-known phenomenon of "manifesting carriers of recessive disease" or the idea that many genes have both a recessive and a dominantly inherited disease association. This is an exceedingly well described phenomenon, especially in cancer predisposition syndromes as well as in X-linked conditions where mosaic X inactivation in females can often lead to variable disease manifestations. In this regard, it would also be relevant to note that the disease phenotype in AR vs AD conditions that arise from the same gene may differ.

- ATM: Recessive Ataxia Telangiectasia is different than dominant ATM-related cancer predisposition

- BRCA2: Recessive Fanconi Anemia is different than dominant BRCA2-related HBOC

- MUTYH: Recessive MUTYH-associated polyposis is different than the statistically increased colon cancer risk in heterozygote carriers

Sometimes this distinction in AR versus AD disease phenotypes in the same gene is due to different disease mechanism (biallelic loss of function versus monoallelic gain of function), but sometimes it is a dosage effect with the phenotypic features either representing dramatically different conditions in the biallelic versus monoallelic state, or sometimes the difference in phenotype is more one of penetrance and expressivity. Therefore, in some cases the phenotypic distinction is such that the conditions caused by monoallelic versus biallelic variants are considered different, whereas in other cases the difference is more of a gradual, dosage effect in which case the term "semi-dominant" is often invoked.

The introduction could therefore acknowledge the existing literature on this subject while indicating that the Finnish biobank is potentially an interesting and powerful way to look for additional such differences (primarily in dosage effect for specific variants).

3. As noted above, it can be disorienting for the reader to toggle between thinking about Mendelian/monogenic versus non-Mendelian/polygenic genotype-phenotype relationships. This is particularly true in Table 2 where the authors examine B/LB variants in GWAS phenotypes (which may or may not match up with the phenotype associated with the Mendelian/monogenic disease association).

Overall, the discussion around B/LB variants is probably a bit overstated. These are simply variants that are asserted to not have a causal effect with respect to a specific monogenic disease. It is not an assertion that they do not play any role in any phenotype. Any confusion on that point is due to individuals misunderstanding the terminology used in classification of variants in monogenic diseases. While one could certainly argue that Mendelian geneticists could have chosen a better term, the discovery of statistically significant PheWAS associations for any given B/LB variant does not negate their status as "benign with respect to causing monogenic disease" but rather indicates what is already known -- that "benign" variants within known "monogenic" disease genes can contribute to polygenic disease risk for complex multifactorial disease. This is well known and should not be particularly surprising.

4. The discussion section is still lacking some important limitations that were raised by the original reviewer. In particular, it may be helpful to indicate that the "phenotype" descriptors that are available via these large biobanks can best be understood as providing a "signal" for individuals who may be penetrant for a monogenic disease. More definitive clinical phenotyping would likely be necessary to establish the clinical correlation between the disease-associated genotype and the symptoms in the manifesting individual.

MINOR POINTS:

1. Figure 2 legend does not match to figure

A = shows curve for cataract (CASP7?)

B = shows curve for hearing loss (C10orf90?)

2. Reference #2 is an interesting choice, given that most cancer geneticists I have interacted with do not think that the phenotype caused by CHEK2 is actually "Li-Fraumeni syndrome" but is actually a misnomer. Consider choosing a different reference to support the assertion that "Rare genetic variants with large effects can have direct implications on potential treatments." (line 55-56)

Author Rebuttals to First Revision:

Referees' comments:

Referee #2 (Remarks to the Author):

The authors have substantially rewritten the manuscript taking account of previous comments from the reviewers. I think it is improved and my specific comments from the first version have been addressed. I do have a few additional comments:

- For genes where you detect an effect in heterozygotes, and have excluded homozygotes, how certain are you that the individuals included are not compound heterozygous? Have you tested this, for example by excluding individuals with >2 non-synonymous variants in the gene?

We computed linkage disequilibrium (LD) between variants in our dataset with a Finnish population reference panel consisting of 2,244 high-coverage (30x) whole genome sequenced individuals (Mitt *et al*, Eur. J. Hum Genet. 2017). We thus determined which pathogenic variants in one gene were in low LD. We excluded individuals with two such pathogenic variants assumed to be likely in compound heterozygous state before testing heterozygous effects in our data. We reference this in Supplementary Note 4 and mention it in the legend of Table 1. However, we cannot exclude all potential compound heterozygous carriers which is a limitation that we discuss in ll. 408 ff.

- I'm unclear of the purpose of table 2 and some further discussion might be useful. If the point is merely to say that being "benign" for a condition in ClinVar doesn't mean a variant is neutral for all other conditions, this is obvious; indeed all assertions of pathogenicity must be linked to phenotypes, and variants can only be benign or pathogenic with respect to a phenotype. However, some clinvar B/LB variants in the table are associated with an increased risk of the same condition. How does this fit with Figure 4? Presumably these are risk alleles that act via a semi-dominant manner but are not causal for monogenic disease? And perhaps it points to the reason that the variant was evaluated by submitters to ClinVar? Also note that at least some of the associations in this table have been seen in previous GWAS.

We agree that the definition of B/LB variants is that they are "not implicated in monogenic disease" however, we think that in reality they are often intuitively considered as neutral. With this analysis we thus wanted to highlight that quite a few B/LB variants are directly causally associated with a disease phenotype, even though the effects are of course much smaller than for a Mendelian condition. We found explanations similar to yours and write a more detailed discussion in Supplementary Note 6 and now clarify also in the main text (ll. 288 ff). We now however moved Table 2 into the Supplement, as we agree, the findings are not very surprising.

- Figure 4 is very helpful, but I wonder if it should be figure 1.

We changed Figure 4 to be Figure 1 and refer to Figure 1 early on the manuscript as an orientation.

You refer to the "homozygous model" and the "additive model" early in the manuscript, but these terms come from very different sections of the genomics community, and it was not immediately clear to me what you meant by comparing these terms. Some of the changes in terminology are to respond to previous

comments from other reviewers, but to me, additive and homozygous are too different to be compared directly; one is a genotype while the other is a mathematical function. Recessive and semi-dominant are better comparative terms, or you could just use homozygous and heterozygous.

We agree with your points and now changed the nomenclature back to 'recessive association' when we refer to the recessive GWAS model. We keep using the term 'homozygous' only when referring to the specific homozygous effect on the phenotype.

For greater consistency with the additive concept, could recessive and dominant be described as threshold models and reduced penetrance and "recessive with rare expressing hets" as multiplicative or exponential models?

We had a longer discussion on how to incorporate your suggestions of the different models. We agree to describe the recessive and dominant model as "threshold models" and included that into our Figure 1.

We are however hesitant in using the terms 'multiplicative' or 'exponential' for models 2 and 4 (reduced penetrance and "recessive with rare expressing hets") for the following reasons. The additive model for a linear regression used in quantitative traits translates to a log-additive (i.e. multiplicative) model in binary trait logistic regression - so for any disease diagnosis the customary GWAS model is in fact multiplicative in risk (that is, if one allele raises risk by 1.5x, 2 copies is expected to elevate risk by 2.25x - NOT 2x). In addition, models 2 and 4 also include a qualitative phenotype aspect (reduced expressivity in heterozygotes), that fits less into a purely quantitative model. We thus feel the pictorial and two explanatory rows already provide enough description for models 2 and 4.

Referee #3 (Remarks to the Author):

The authors have revised the paper to address some of the previous comments. I still found it quite hard to follow because of carelessness in how figures are ordered and referenced, unnecessary acronyms, and disordered explanations of unusual terms. All of these can be fixed (I enumerate some below), but they made the review unnecessarily hard going.

There's also a lot of information consigned to supplementary notes. I appreciate the difficulty of putting many results into length limits etc, and the previous review asked for a streamlined main message. But some of them (e.g. supplementary note 6) are mini papers in themselves with results, methods and discussion.

I also have two substantive science comments:

1. On the GWAS analysis, it is interesting that homozygote effects are found that are missed by a standard additive test, but studies from a long time ago, such as the 2007 WTCCC paper, fit multiple models of inheritance in their scans. Of course, in the intervening years the additive model has been such a good fit for most common GWAS associations, that this is no longer widely done. So this is a timely reminder to revisit the topic, but not really a new idea.

Absolutely agree - in the introduction, we now give some context on what has been previously done on the topic of non-additive GWAS models as well as on inheritance models in Mendelian genetics. Indeed the WTCCC was such a model that the failure

of the additional degree of freedom to turn up additional findings (in retrospect much more attributable to how little we appreciated the sample size required for GWAS discovery) was used as a point of reference in favor of only using the additive model.

That paper (and others) have also used more formal approaches to evaluate the relative fit of recessive vs additive (or 2df) models, which would be better to use than the "2 orders of magnitude p-value" rule of thumb used here.

We intensely discussed whether to use a classic 2 d.f. model or directly investigating homozygous effects and chose the latter for better interpretability in clinical genetics settings and straightforward implementation within our SAIGE GWAS model required for robust and scalable analysis. This model fulfilled our goal to identify large-effect recessive associations, it does not have optimal power for evaluating small deviations from the additive model. In addition, we were not fully convinced that modest deviations in favor of the 2 d.f. model over the additive model might not also arise from data accuracy or completeness differences that vary by genotype though agree this is an important aspect to investigate given the current results.

Thus, we conducted and have added additional simulations to further understand the output of our recessive and additive model in the presence of a strictly recessive or additive association (for details see Supplementary Figures 3, Supplementary Note 2). Here, we find that if a simulated recessive effect (heterozygous effect 1, homozygous effect 5, 100,000 simulations at minor allele frequency thresholds of 0.01, 0.02, 0.05 and 0.1), reaches genomewide significance in the recessive model (recessive p-value $<5 \times 10^{-8}$), the recessive p-value is 2 orders of magnitude below the additive p-value in 298,414/298,418 instances across 4x100,000 simulations. Thus, if truly recessive associations reach genomewide significance, we will likely identify them as recessive with our approach.

Conversely, we find that for a simulated strictly additive effect in logarithmic space (heterozygous effect 1.5, homozygous effect 2.25; 100,000 simulations each at minor allele frequency thresholds 0.01, 0.02, 0.05, 0.1) recessive p-value is 2 orders of magnitude below the additive p-value in 0/400,000 simulations. We can thus with high confidence reject strict additivity when recessive p-value is 2 orders of magnitude below the additive p-value.

2. I think the DBH presentation isn't quite right. Its in a section about finding errors in causal variants for monogenic disease. And the variant here is called "probably causal". But probably causal for what? It seems highly unlikely that it is causal for the Mendelian DBH deficiency syndrome, which is what is implied. That syndrome is characterised by complete absence of the DBH enzyme, and this 2011 paper says there are 15 known patients worldwide (<https://www.nature.com/articles/npp201142>). So it seems much more likely that this variant does affect the function of DBH, and thus yields modest lifetime protection from hypertension, but I don't think it should be re-classified as a pathogenic variant in OMIM.

Thanks for catching this miswording! By 'probably causal' we referred to the statistical finemapping results (posterior inclusion probability >0.99), not to the Mendelian phenotype, but we appreciate now how that could cause confusion in that paragraph! So we did not intend that this variant should be re-classified as pathogenic but rather agree with your interpretation and tried to make this now clearer in the manuscript (ll. 288 ff). We think this is a great example of an allelic

series where one variant has a milder effect on a similar phenotype as the phenotype caused by pathogenic variants in the same gene.

And I suspect this same concept may well play in some of the other cases of benign variants with a detected effect on a related phenotype in the EHRs (e.g. the 16 candidate causal SNPs from supplementary note 6). And isn't this expected from lots of previously published results where GWAS hits are more overlapping than chance with Mendelian genes for related phenotypes? I think it is likely (or certainly possible) that many of these "benign" variants are not neutral, but also that they are not pathogenic in the clinical genetics sense.

Yes, we give a similar explanation also in Supplementary Note 6 and now clarified a bit more in the main text. We now moved Table 2 into the Supplement since we agree this is not a particularly surprising observation.

Minor:

B/LB and P/LP are unhelpful acronyms. I think it would be better to just use benign and pathogenic, or "likely benign" and "likely pathogenic". I appreciate that the labels are different in OMIM, but for me the cognitive load of parsing them was much higher than remembering that you were including two levels of confidence in the analysis. We changed it to "likely benign" and "likely pathogenic" close to our definition of "(likely) benign" and "(likely) pathogenic" in the first version of the manuscript.

Figure 3C is referred to before Figure 2 (or the rest of Figure 3), then Figure 3B is referred to before Figure 3A.

We tried to refer to all Figures in the correct order.

NFSEE is defined the second time it is used, rather than the first
We fixed this.

"comp-het" is not a necessary shortening of "compound heterozygous", and is only used once later in the paper anyway

We agree that using the full expression is easier to follow and thus changed.

Figure 4 is referenced before all the other figures, because it is framed as a summation in the discussion, but also helpful to refer to earlier, but the order has to be resolved one way or the other.

We changed the order now making Figure 4 to Figure 1.

"FinnGen R6 data" is discussed without really defining what it is. I gather it is an update to the dataset since the initial submission?

It is indeed. We now better introduce it in the results (I. 193) and also give a bit more detail in the Methods (II. 629 ff).

Referee #4 (Remarks to the Author):

Heyne and colleagues have utilized Finnish biobank data to examine "Mendelian disease architecture" utilizing the unique population history of the Finnish population to examine bottleneck events that created enrichment for certain variants. They

propose to examine homozygous and heterozygous effects of >40,000 coding variants on >2000 disease phenotypes using the EHR data for 176,899 individuals. As a reviewer who did not participate in the first round of review, I am asked to assess whether the authors have adequately responded to a previous review. I am also asked to provide an independent critique of the overall manuscript.

Overall, I like the idea of using "disease architecture" in the title, but I wonder if a more general phrasing of "architecture of genetic disease" would be more appropriate, given how this work blends GWAS/quantitative genetics with the types of genetic variation that underlies Mendelian/monogenic phenotypes.

We adopted that suggestion and changed the title accordingly.

One of the main concerns of a previous reviewer was in the use of language to describe inheritance patterns, zygosity, allelic requirement, and disease phenotype. This is in part challenging because of the dual focus on quantitative trait/multifactorial/polygenic disease genetics and Mendelian/monogenic conditions. It can indeed be quite disorienting to toggle between thinking about variants in a "Mendelian" framework where the majority of the phenotype can be attributed to the underlying genotype, versus quantitative genetics in which small polygenic effects contribute as an ensemble. Overall, it appears that the authors have done a good job of addressing this concern, aside from a few instances in which additional clarity may be needed.

On the whole, this manuscript is quite interesting and demonstrates the utility of isolated populations to uncover details of the complexity of genetic influence on disease (which, of course, we have long recognized can cross the traditional divide between polygenic and monogenic).

SPECIFIC COMMENTS:

1. The summary paragraph still includes some slight confusion about the use of term "recessive" which in Mendelian terms should be applied to observed phenotypes (attributable, of course, to biallelic variants). I would simply suggest removing "recessively inherited" from the first sentence. The next sentence would then adequately indicate that the study is interested in homozygous variants that influence recessive diseases, with the slight modification of something like "facilitates the identification of biallelic variants that cause recessive disease phenotypes."

We incorporated your suggestions into our introductory sentences - thank you!

2. The introduction needs to acknowledge the well-known phenomenon of "manifesting carriers of recessive disease" or the idea that many genes have both a recessive and a dominantly inherited disease association. This is an exceedingly well described phenomenon, especially in cancer predisposition syndromes as well as in X-linked conditions where mosaic X inactivation in females can often lead to variable disease manifestations. In this regard, it would also be relevant to note that the disease phenotype in AR vs AD conditions that arise from the same gene may differ.

- ATM: Recessive Ataxia Telangiectasia is different than dominant ATM-related cancer predisposition

- BRCA2: Recessive Fanconi Anemia is different than dominant BRCA2-related HBOC

- MUTYH: Recessive MUTYH-associated polyposis is different than the statistically increased colon cancer risk in heterozygote carriers

Sometimes this distinction in AR versus AD disease phenotypes in the same gene is due to different disease mechanism (biallelic loss of function versus monoallelic gain of function), but sometimes it is a dosage effect with the phenotypic features either representing dramatically different conditions in the biallelic versus monoallelic state, or sometimes the difference in phenotype is more one of penetrance and expressivity. Therefore, in some cases the phenotypic distinction is such that the conditions caused by monoallelic versus biallelic variants are considered different, whereas in other cases the difference is more of a gradual, dosage effect in which case the term "semi-dominant" is often invoked.

The introduction could therefore acknowledge the existing literature on this subject while indicating that the Finnish biobank is potentially an interesting and powerful way to look for additional such differences (primarily in dosage effect for specific variants).

This is a balanced description of the topic that we tried to include as space permitted in the introduction. We now give a brief introduction to the 2 related main questions of

- 1) Mendelian inheritance pattern beyond recessive/dominant as well as
- 2) non-additive GWAS.

We had previously included a statement about the sometimes-different molecular effects for differently inherited diseases quoting the example of SERPINA1 and Alpha1 antitrypsin deficiency (loss of function: lung disease, gain of function: liver disease). We earlier removed it due to issues of space but now include it in the introduction.

3. As noted above, it can be disorienting for the reader to toggle between thinking about Mendelian/monogenic versus non-Mendelian/polygenic genotype-phenotype relationships. This is particularly true in Table 2 where the authors examine B/LB variants in GWAS phenotypes (which may or may not match up with the phenotype associated with the Mendelian/monogenic disease association.

Overall, the discussion around B/LB variants is probably a bit overstated. These are simply variants that are asserted to not have a causal effect with respect to a specific monogenic disease. It is not an assertion that they do not play any role in any phenotype. Any confusion on that point is due to individuals misunderstanding the terminology used in classification of variants in monogenic diseases. While one could certainly argue that Mendelian geneticists could have chosen a better term, the discovery of statistically significant PheWAS associations for any given B/LB variant does not negate their status as "benign with respect to causing monogenic disease" but rather indicates what is already known -- that "benign" variants within known "monogenic" disease genes can contribute to polygenic disease risk for complex multifactorial disease. This is well known and should not be particularly surprising. While we fully agree to this definition of benign variants we argue that human geneticists are less aware that benign variants can have a phenotypic effect. (There is some literature where benign variants are used as 'neutral' variants and we also received this feedback from several geneticists.) We explained this also in more

detail in a Supplementary Note 6. We now however moved the Table 2 into the Supplement since we agree this is per se not a particularly surprising observation.

4. The discussion section is still lacking some important limitations that were raised by the original reviewer. In particular, it may be helpful to indicate that the "phenotype" descriptors that are available via these large biobanks can best be understood as providing a "signal" for individuals who may be penetrant for a monogenic disease. More definitive clinical phenotyping would likely be necessary to establish the clinical correlation between the disease-associated genotype and the symptoms in the manifesting individual.

We now expanded on this limitation in the discussion (see ll. 414 ff.).

MINOR POINTS:

1. Figure 2 legend does not match to figure
A = shows curve for cataract (CASP7?)
B = shows curve for hearing loss (C10orf90?)

We fixed this.

2. Reference #2 is an interesting choice, given that most cancer geneticists I have interacted with do not think that the phenotype caused by CHEK2 is actually "Li-Fraumeni syndrome" but is actually a misnomer. Consider choosing a different reference to support the assertion that "Rare genetic variants with large effects can have direct implications on potential treatments." (line 55-56)

Thank you for catching that! We chose a different reference at that place.

Reviewer Reports on the Second Revision:

Referees' comments:

Referee #2 (Remarks to the Author):

The authors have addressed all my comments - and those from other reviewers - very well and I have no further comments.

Referee #4 (Remarks to the Author):

I appreciate the efforts the authors have made to respond to the extensive comments! This is a challenging topic and I think they have improved the clarity of discussion of allelic requirements and genotype/phenotype associations.